# Direct in situ photolithography of perovskite quantum dots based on photocatalysis of lead bromide complexes

Pingping Zhang[1], Gaoling Yang [2] ✉, Fei Li[3], Jianbing Shi[1] & Haizheng Zhong [1]

Photolithography has shown great potential in patterning solution-processed nanomaterials for integration into advanced optoelectronic devices. However, photolithography of perovskite quantum dots (PQDs) has so far been hindered by the incompatibility of perovskite with traditional optical lithography processes where lots of solvents and high-energy ultraviolet (UV) light exposure are required. Herein, we report a direct in situ photolithography technique to pattern PQDs based on the photopolymerization catalyzed by lead bromide complexes. By combining direct photolithography with in situ fabrication of PQDs, this method allows to directly photolithograph perovskite precursors, avoiding the complicated lift-off processes and the destruction of PQDs by solvents or high-energy UV light, as PQDs are produced after lithography exposure. We further demonstrate that the thiol-ene free-radical photopolymerization is catalyzed by lead bromide complexes in the perovskite precursor solution, while no external initiators or catalysts are needed. Using direct in situ photolithography, PQD patterns with high resolution up to 2450 pixels per inch (PPI), excellent fluorescence uniformity, and good stability, are successfully demonstrated. This work opens an avenue for non-destructive direct photolithography of high-efficiency light-emitting PQDs, and potentially expands their application in various integrated optoelectronic devices.

Perovskite quantum dots (PQDs) exhibit narrow photoluminescence (PL) emission with high quantum yields which is tunable over the entire visible range, have shown great potential in a range of optoelectronic applications[1–4]. In particular, there is intense interest in applying PQDs in displays[5–8]. Despite the great progress, PQDs based light-emitting diodes (LEDs) still suffer from some issues that prevent their incorporation into commercial display products such as cell phones or laptops. One of them is patterning, multicolor LEDs have been hindered by the difficulty in patterning red, green, and blue (RGB) PQD films into arrays of individual devices rather than bulk films[9–12]. Therefore, the capability of patterning PQDs into pixel arrays is urgently needed.

Over the past few years, various patterning techniques have been developed, including inkjet printing[13], nanoimprinting[14], laser direct writing[15], photolithography[16], and so on. Among all these methods, photolithography has attracted considerable attention owing to its high resolution, wide availability, and high throughput. There are two categories of this technique, conventional photoresist-assisted method, and direct photolithography. The former uses the conventional lift-off technique where the photoresist is first patterned before QDs are deposited[17–20]. However, PQDs are easily dissolved in polar solvents and re-dispersed in nonpolar solvents due to their ionic nature, making them usually incompatible with traditional photolithography processes

[1]MIIT Key Laboratory for Low Dimensional Quantum Structure and Devices, School of Materials Science and Engineering, Beijing Institute of Technology, Beijing 100081, China. [2]MIIT Key Laboratory for Low Dimensional Quantum Structure and Devices, School of Optics and Photonics, Beijing Institute of Technology, Beijing 100081, China. [3]QD LAB, Hefei Innovation Research Institute of Beihang University, Hefei, Anhui 230001, China. ✉e-mail: glyang@bit.edu.cn

where a lot of solvents are needed in photoresists, developers, etchants, or strippers[11]. Alternatively, direct photolithography is quite simple and effective, which patterns photosensitive QDs directly through the light-induced change in their solubility[21]. Recently, several research groups presented high-resolution QD patterns based on direct photolithography with different photochemistry mechanisms, such as unsaturated double bond cross-linking[22–24], azide cross-linking[24–27], polarity change[28–30], and so on.

Most of these photolithography technologies are based on photopolymerization, which is the curing process of monomers triggered by photosensitizers or photoinitiators exposed to light. Recently, perovskite has shown great promise as an efficient catalyst[31,32], and is used to directly initiate the photopolymerization process for the fabrication of various PQD-polymer compounds, such as PQD-polymethyl methacrylate[33], PQD-polystyrene[34], and PQD-polyaniline[35]. However, these photopolymerizations can be easily inhibited in the air due to the quenching effect of oxygen on carbon radicals[36]. While the thiol-ene reaction is insensible to oxygen and high-efficiency, which has been applied in photolithography for microdevice fabrication[37]. A typical radical thiol-ene reaction often starts from thiyl radicals, then they attack double bonds to generate carbon radicals, which in turn react with thiol monomers to release thiyl radicals, resulting in the chain growth and transfer[38]. The insensitivity to oxygen comes from that the ready hydrogen abstraction of peroxy radicals from thiols to generate active thiyl radicals, which have the ability to continue the free-radical chain process[39]. Thioethers have been photosynthesized with PQDs as a photocatalyst[40]. However, most reactions with PQDs as a photocatalyst normally involve blue or white light irradiation for a long time of several hours which may also potentially destroy the PQDs[41] and are hard to incorporate existing photolithography process. Hence, PQDs may not be the best option for photocatalysis during the existing photolithography process.

Additionally, in the existing direct photolithography, almost all photosensitive QDs are prepared through the hot injection approach and subsequent ligand exchange method beforehand, which usually involve high reaction temperature, harsh inert gas environment, complex synthesis steps, and extensive cleaning steps. In addition, these photosensitive QDs suffer from preserving their optical properties in the lithography exposure process, especially for PQDs. In situ fabrication has been proved to be a powerful method to synthesize PQDs directly on a substrate or in a matrix[42], including polymer matrix[43], glass matrix[44], molecular sieves[45], or crystals[46], due to the low formation enthalpy and high defect tolerant of ionic perovskite[47,48]. Recently, in situ fabrication has been utilized to provide patterned PQDs pixels with tunable luminescence for integrating into optoelectronic devices[49,50].

Thus motivated, here we propose a direct photolithography method to pattern in situ fabricated PQDs based on polymerization catalyzed by lead bromide complexes. This method produces high-resolution patterns by direct photolithography of perovskite precursors using UV light to trigger the photopolymerization, PQDs are in situ fabricated in polymer by final annealing, avoiding high-energy UV light to destroy PQDs since they were produced after exposure. In particular, we demonstrate that the photopolymerization under UV irradiation is based on lead bromide complexes catalyzing the thiol-ene free-radical reaction, without any external initiators or catalysts that are harmful to the stability of patterns. Furthermore, uniform and residual-free luminescent pixel arrays are fabricated via direct photolithography of in situ fabricated PQDs, in which the high uniformity of emission intensities reflects the homogeneous distribution of PQDs. Lastly, we successfully fabricate red, green and blue PQD patterns, the minimum feature size of 5 μm corresponding to a resolution of up to 2450 pixels per inch (PPI), the maximal thickness of 10 μm enables to absorb blue or UV light more efficiently when used as color convention pixels. The direct photolithography method combined with in situ synthesis strategy, clearly provides an efficient platform for the manufacture of high-resolution patterned optoelectronic devices.

## Results

### Description of direct in situ photolithography method

Fine PQDs patterns were fabricated from the original perovskite precursor solution with monomers by direct in situ photolithography method. As shown in Fig. 1a, prior to photolithography, the glass substrate is cleaned by sonication in various solvents, then silane coupling agents are used to functionalize the substrate with ethenyl or thiol groups exposing. Such functionalized substrate creates strong covalent bonding sites for the resultant polymer film, which is critical to the successful patterning of perovskite films. The patterning of PQDs can start from simply casting the photosensitive perovskite precursor resist (PPR) directly onto the functionalized substrate. The PPR is key to the direct in situ photolithography, which is prepared by dissolving reagent salts, multifunctional thiol and ethenyl monomers in polar aprotic solvents. Specifically, for green PPR, MABr (MA = methylammonium) and PbBr$_2$ are chosen as perovskite precursor salts, which have been proved to be very effective reagents to generate perovskite. For monomers, trimethylolpropane tris (3-mercaptopropionate) (TTMP), a multi-thiol crosslinking reagent, is used as a thiol monomer, while triallyl isocyanurate (TAIC) provides electron-rich vinyl groups. To dissolve perovskite precursor and monomers better, polar aprotic solvents such as N,N-dimethylformamide (DMF) and dimethyl sulfoxide (DMSO) are the best choice. Unlike previous methods, direct in situ photolithography approach directly patterns original perovskite precursors instead of prepared QD inks, which often need extensive cleaning steps and can be destroyed under high-energy UV light in the presence of oxygen[51,52]. Second, perovskite precursor film is exposed with UV light through a photomask. Upon UV irradiation, a photochemically activated reaction between TTMP and TAIC occurs, leading to the solidification of the PPR in the exposed area. Third, the unexposed PPR is removed by spin-washing with chloroform as a developer. Cured products on exposed areas can adhere to the substrate strongly during spin-washing because of the covalent interactions formed between functionalized substrates and TTMP or TAIC in PPRs, making the developing process much easier. Finally, the prepared perovskite precursor patterns are directly annealed in the ambient air to evaporate residual solvents, when the concentration reaches its critical value for in situ nucleation, luminescent perovskite patterns are formed. Figure 1b shows the corresponding fluorescence image of the patterned arrays of square with a size of 30 μm on a substrate, the patterned pixel arrays emit bright and uniform green fluorescence under UV irradiation. Cross-section transmission electron microscopy (TEM) measurement was performed to clearly demonstrate the QD structure of perovskite, and exhibit their homogeneous dispersion in micropatterns, as shown in Fig. 1c. High crystallinity and continuous lattice fringes can be seen clearly from further high-resolution TEM (HRTEM) image in Fig. 1d, the lattice constant of 0.301 nm is consistent with (002) interplanar spacing of MAPbBr$_3$. The bright and uniform PQD-polymer films can be ascribed to the good protection of polymer matrix on the exposed region, which was proved by the calculation of the retention of PQDs defined as the percentage of PQDs retained in the exposed films after developing[24]. Inductively coupled plasma-optical emission spectroscopy (ICP-OES) and inductively coupled plasma-mass spectroscopy (ICP-MS) were used for the analysis of Pb atom in the developer solvent and polymer film to determine the retention of PQDs. And the figure was recorded as up to 85%, which accounts for the good protection effect of polymer on perovskite thus obtaining high luminescent patterns.

### Exploration of photochemical mechanism

In general, external initiators or catalysts are essential for polymerization, but things are different in our case. It seems perovskite precursors have the ability of photocatalysis, which makes direct

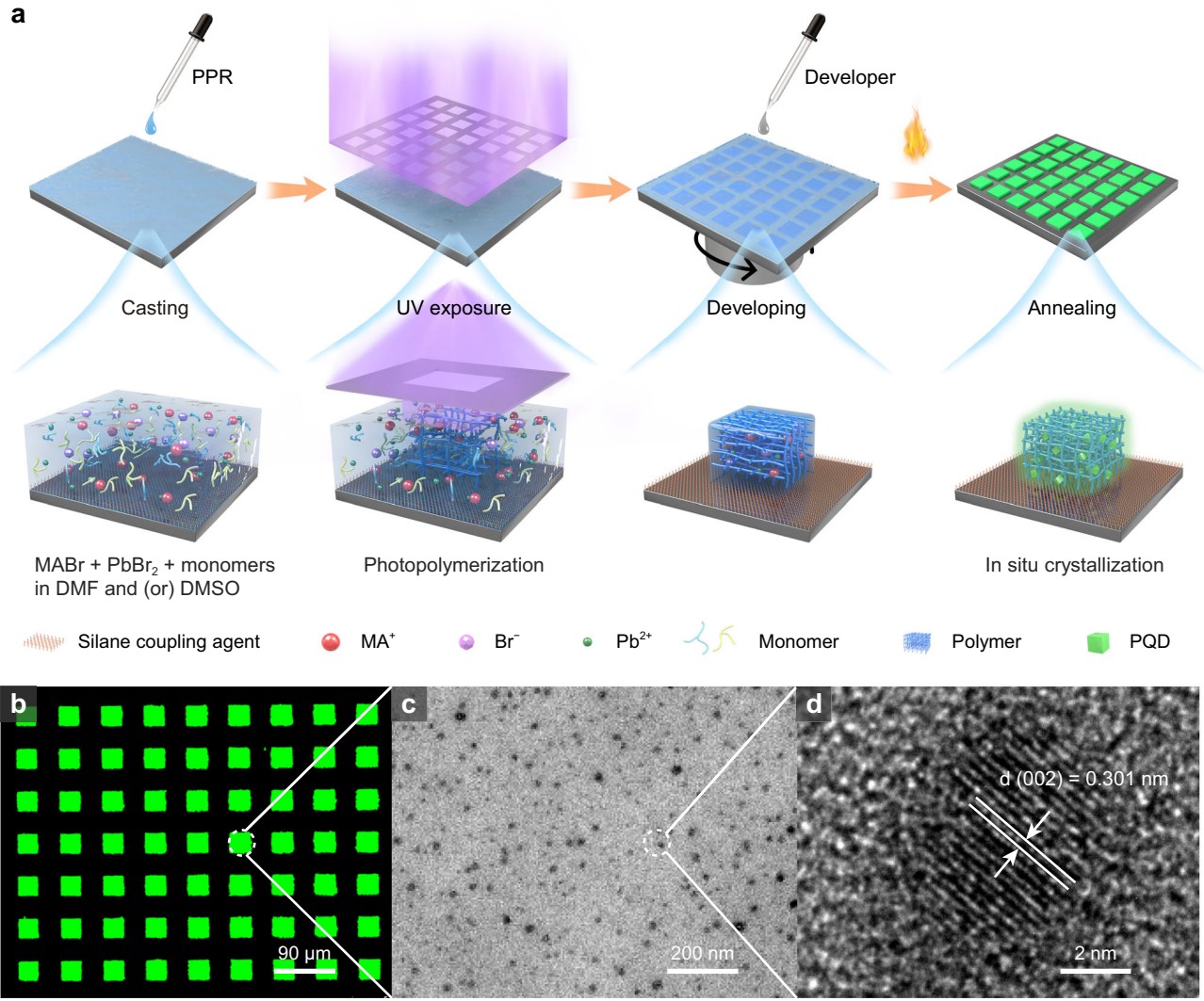

**Fig. 1 | Schematic illustration of PQD patterning method. a** Schematic description of the direct in situ photolithography method from the perovskite precursor resist (PPR), annealing refers to heating the samples at a specific temperature. **b** Fluorescence image of green MAPbBr$_3$ PQD patterns (30 µm size square) fabricated by the direct in situ photolithography method. **c** Cross-section TEM image of MAPbBr$_3$ PQD patterns fabricated by the direct in situ photolithography method. **d** HRTEM image of single MAPbBr$_3$ PQD in the patterns fabricated by the direct in situ photolithography method.

in situ photolithography of PQDs possible. We seek to understand the photocatalysis mechanism by investigating the type of polymerization, photocatalytic species, and reactive site. To verify the photocatalysis of perovskite precursors, equal volumes of monomer systems with and without perovskite precursors were irradiated under 365 nm UV light for 2 min. As can be seen in Fig. 2a, with perovskite precursors, solidified gelatin was formed at the bottom of the bottle, while without precursors, the ink was still liquid, appreciably illustrating the photocatalysis of perovskite precursors. To further explore whether the photopolymerization was conducted by self-polymerization[31,33] or copolymerization[37], monomer systems either with one single monomer (ethenyl or thiol) or with both of them mixing with perovskite precursor were irradiated under UV light. Evidently, the monomer system containing both ethenyl and thiol monomers was cured within 30 s (entry 1 in Supplementary Table 1 and Supplementary Fig. 1), while for a single monomer, no polymerization occurred (entries 2 and 3 in Supplementary Table 1 and Supplementary Fig. 1). This indicates that the PPR underwent the thiol-ene reaction. Additionally, this reaction system was further verified by Fourier-transform infrared (FT-IR) spectra of the PPR before and after UV irradiation, both peaks of −CH=CH$_2$ (3080 cm$^{-1}$) and −SH (2527 cm$^{-1}$) reduced dramatically after

100 s irradiation (Fig. 2b), confirming the thiol-ene photopolymerization system. The extent of polymerization was also determined by FT-IR spectra, where 83% and 84% were ascribed to the conversion rate of the ethenyl and thiol group, suggesting the nearly 1:1 stoichiometry rection (Supplementary Fig. 2). Furthermore, various monomers with varying quantities of functional groups (Supplementary Fig. 3a, b) were also examined under the same reaction conditions, all the combinations were cured within 30 min under a low power UV light (10 mW cm$^{-2}$) (Supplementary Fig. 3c and Supplementary Table 2), demonstrating the universality of thiol-ene reactions photocatalyzed by perovskite precursors.

Generally, the thiol-ene reaction has two polymerization mechanisms: thiol-Michael addition which is catalyzed mainly by bases or nucleophiles[53], and thiol-ene free-radical addition which is induced by photoinitiators or high-energy UV light irradiation[37]. To unravel the polymerization mechanism of the PPRs, proton source and free-radical inhibitors were added, which can effectively inhibit the thiol-Michael addition and free-radical addition, respectively. When a proton source such as HCl was added in the PPR, the cured polymer was obtained after 19 s irradiation using a 365 nm UV light source (entries 5 vs 4 in Supplementary Table 3 and Supplementary Fig. 4), implying this

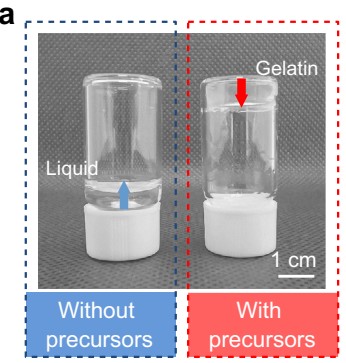
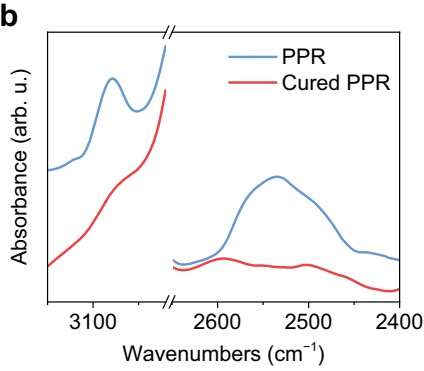
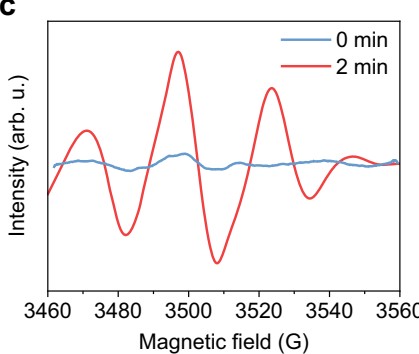

**Fig. 2 | Investigation of the photopolymerization type. a** Photograph of the resultant inks without (left side) and with (right side) perovskite precursors in the monomer system after irradiating with 365 nm UV light (20 W) for 2 min. **b** FT-IR spectra of the perovskite precursor resist (PPR) and cured PPR after 365 nm UV light (5 W) irradiation for 100 s. **c** EPR spectra of the PPR collected after 0 min and 2 min UV irradiation. Source data are provided as a Source Data file.

system is not the thiol-Michael addition mechanism. When a negligible amount of 5,5-dimethyl-1-pyrroline N-oxide (DMPO) was used as a free-radical inhibitor, no polymerization happened under the same UV light environment (entries 6 vs 4 in Supplementary Table 3 and Supplementary Fig. 4), confirming the inhibiting effect of the free-radical inhibitor for this reaction system. To further confirm the presence of free radicals, electron paramagnetic resonance (EPR) spectroscopy measurement was carried out. A clear EPR signal assigned to carbon radicals was detected after 2 min irradiation of PPR using a UV light source (Fig. 2c), unambiguously proving the growth and transfer of polymer chain was initiated by free radicals. Together, both the inhibition reaction and EPR results strongly confirm the free-radical reaction pathway.

We were intrigued by the unexpected photocatalytic ability of perovskite precursors and curious about which part plays a major role. Firstly, to explore whether the reagent salts MABr and $PbBr_2$ were the primary catalyst, a series of control experiments were performed with various constituents. We found that nearly all the liquid was cured in 6 min when MABr and $PbBr_2$ were used together (entry 7 in Supplementary Table 4 and Supplementary Fig. 5a), reaction conducted using $PbBr_2$ also gave cured products but at a low yield after 20 min irradiation (entry 8 in Supplementary Table 4 and Supplementary Fig. 5b), no cured product appeared when MABr was used (entry 9 in Supplementary Table 4 and Supplementary Fig. 5b). This result implies the fast catalytic effect of the mixture of MABr and $PbBr_2$, the slow catalytic effect of $PbBr_2$, and excludes the catalytic possibility of MABr, $MA^+$ and $Br^-$ in this system. Obviously, $PbBr_2$ does not play a major catalytic role in this photocatalytic reaction, but it does show some catalytic effect, so we wondered if $Pb^{2+}$ might be the catalyst. Therefore, we examined its catalytic ability by changing the halides from Br to I and Cl. However, no cured product was obtained under UV light (311, 365, and 405 nm) irradiation (entries 11 and 12 vs 10 in Supplementary Table 5, Supplementary Fig. 6), excluding the catalytic effect of $Pb^{2+}$ either. Since the PQDs have been reported to be used as photocatalyst for thiol-ene reactions under blue light irradiation[40], it is easy to relate the photocatalytic effect to the PQDs. However, both the PL and ultraviolet-visible (UV–Vis) spectra (Supplementary Fig. 7) of the PPR exhibit distinctly different emission and large Stokes shift compared with the typical PQDs, demonstrating that there was no PQD generated from PPR no matter how long the solution was stirred. All the PPRs with different stirring time can be cured within 2 min (the left five in Supplementary Fig. 8), while no cured product was found in the ink using PQDs as photoinitiator (the rightest in Supplementary Fig. 8), which at least indicates that PQDs cannot photocatalyze the polymerization as efficiently as the perovskite precursors. Furthermore, MABr was substituted by HBr to eliminate A site ion that is necessary for the fabrication of perovskite, avoiding any possibility to produce perovskite.

The ink with HBr had a comparable curing rate to that with MABr (Supplementary Fig. 9), along with the photocatalysis capacity of $PbBr_2$ demonstrated before, which absolutely excluded the photocatalysis effect of PQDs in our case. These results suggest that the primary catalyst of this reaction is neither from any single reagent salts and their constituent ions in the perovskite precursor solution nor PQDs.

Perovskite precursor solutions are actually colloid[54], where halides and solvent molecules coordinate with lead ions to obtain a variety of lead halide complexes $PbX_n^{2-n}$ (X = Cl, Br or I; $n$ = the coordination number of halide), all the various complexes are in dynamic equilibrium[55]. Therefore, we speculated that the photopolymerization in this system may be catalyzed by the lead bromide complexes. To confirm the catalytic capacity of lead bromide complexes, PPRs with different bromide concentrations were tested. As can be seen in Fig. 3a, with the increase of [MABr], the curing time dramatically reduced, with a maximum increase in catalytic effect by a factor of ten. Since high coordinated lead bromide complexes are formed at high bromide concentration, which can be confirmed by the redshift of absorption (Supplementary Fig. 10a)[56], the enhanced catalytic effect at high [MABr] can imply the stronger catalytic capacity of high coordinated lead bromide complexes. To further demonstrate that, perovskite precursors were dissolved in solvents with different coordination abilities (DMSO > DMF > GBL (γ-butyrolactone))[55]. As solvent molecules can compete with halides for coordination with lead ions, when the coordination ability of a solvent was reduced, the coordination of halide and $Pb^{2+}$ would increase to form lead halide complexes with high coordination numbers[57], which can be proved by the redshift of their absorption spectra (Supplementary Fig. 10b). As shown in Fig. 3b, when the curing yields after 2 min 30 s UV light irradiation were plotted against the decreasing coordination ability of solvents, a clear upward trend emerged (0% for DMSO, 20% for DMF, 70% for complex DMF/GLB), strongly suggesting the key photocatalytic role of high coordinated lead bromide complexes. In perovskite precursor solutions, $PbBr_2$, $PbBr_3^-$, $PbBr_4^{2-}$ have been reported in the literature[56], while $PbBr_2$ has been excluded in the previous experiment, we speculated that high coordinated $PbBr_4^{2-}$ is the key photocatalyst in this reaction. To prove that, UV lights with various wavelengths were used, since lead bromide complexes with different coordination numbers show different absorption of light at different wavelengths[56,58]. As shown in Fig. 3c, when exposed under 311 nm UV light close to the absorption peak of $PbBr_3^-$, an acceptable exposure dose for completely curing were required, combining with the absolutely predominated concentration of $PbBr_3^-$ in the PPR, indicating a relatively weak catalytic effect of $PbBr_3^-$. When replacing with 365 nm UV light close to the absorption peak of $PbBr_4^{2-}$, only half exposure dose was needed even under a relatively low concentration, implying the strong photocatalytic capacity of $PbBr_4^{2-}$. At 405 nm, the

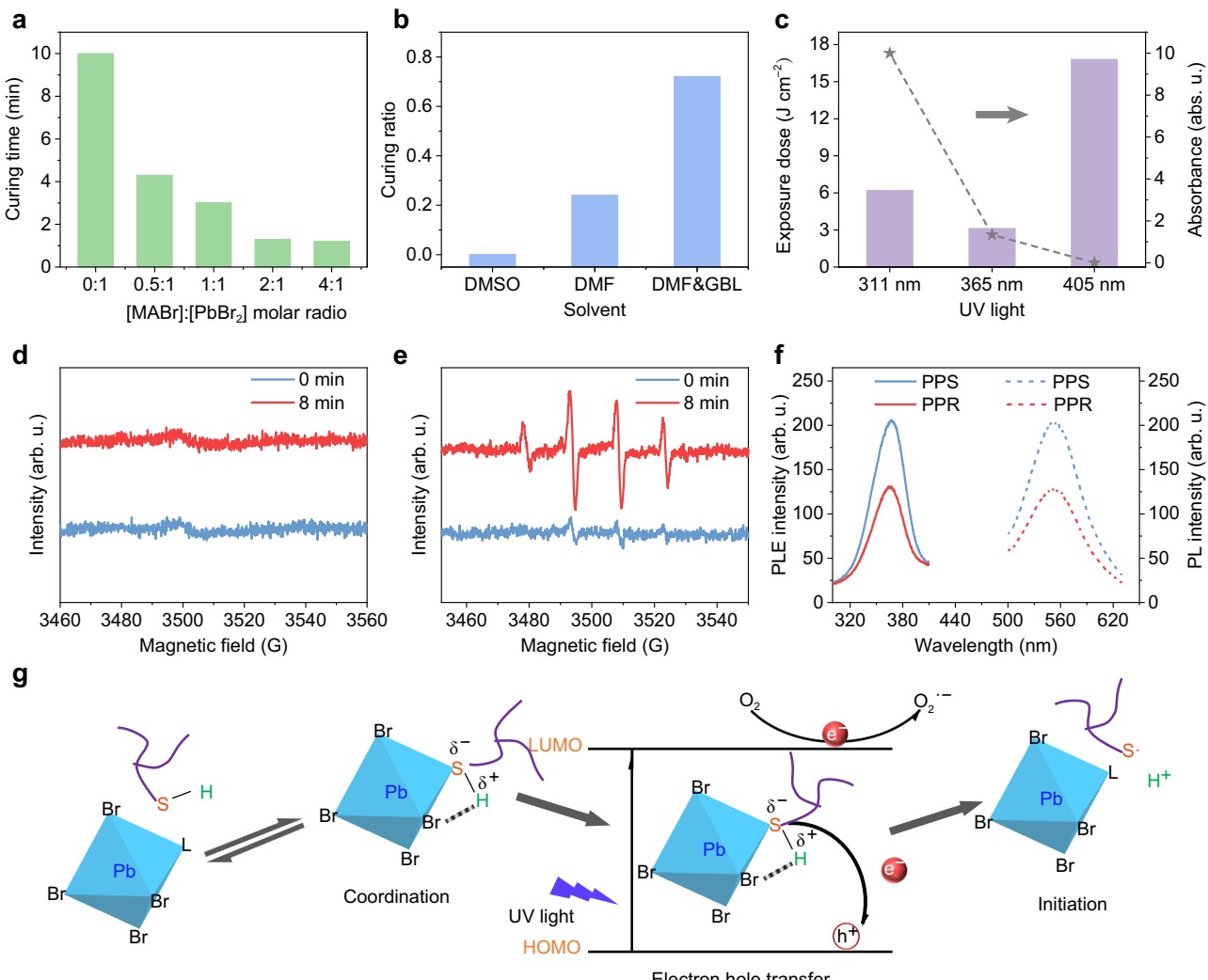

**Fig. 3 | Exploration of the photochemical mechanism. a** Curing time of perovskite precursor resists (PPRs) with an increasing ratio of MABr to PbBr$_2$ at a constant concentration of PbBr$_2$. **b** Curing yield of PPRs with different solvents at constant concentrations of all reagent salts and monomers. **c** Exposure dose of UV lights with different wavelengths to cure the PPR completely (histogram corresponding to the left axis) and absorbance of the PPR at different UV light wavelengths (line graph corresponding to the right axis). EPR spectra of the inks respectively mixed by **d** TAIC and **e** TTMP with the same perovskite precursor solution. **f** PLE (solid line) and PL (dot-dash line) spectra of the perovskite precursor solution (PPS) and PPR. **g** Proposed photochemical mechanism of lead bromide complexes catalyzed initiation process of thiol-ene free-radical addition, L stands for a solvent molecule. Source data are provided with this paper.

absorbance is nearly 0, indicating the low concentration of higher coordinated lead bromide complexes. Based on these results, it can be concluded that the highly coordinated lead bromide complex has a strong catalytic capacity, while in our case, it is PbBr$_4^{2-}$.

To gain further insight into the photopolymerization mechanism, several experiments were conducted to determine the reactive site. We first added different kinds of monomers into the perovskite precursor solution separately (entries 13 and 14 in Supplementary Table 6), where pH went down evidently only after TTMP was added. Meanwhile, a significant decrease in pH can be seen when PbBr$_2$ was added to the monomer system (entry 15 in Supplementary Table 6). The increased acidity of the solutions may come from the weakening of sulfur–hydrogen bonding, indicating the effective coordination between sulfur in thiol groups and lead ions. To further verify this statement, oleic acid (OA), which can coordinate with lead ions effectively was added into the PPR[59]. Almost double curing time can be explained by the reduction of coordination between sulfur and lead ions, verifying the reactive site is on the lead ion (entries 16 and 17 in Supplementary Table 7). Moreover, strong sulfur radical signals were detected after 8 min irradiation when TTMP was added separately into

the perovskite precursor solution (Fig. 3e), while no carbon radical was detected when TAIC was used (Fig. 3d). The EPR measurements suggest that the monomers catalyzed by lead bromide complexes are thiols. Together, all these results confirm that the sulfur–lead coordination bond is the reactive site.

Furthermore, fluorescence quenching studies were performed to prove the electron–hole transfer between PbBr$_4^{2-}$ and organic monomers. As shown in Fig. 3f, both the intensity of photoluminescence excitation (PLE) and PL spectra of the perovskite precursor solution drop more than 35% with limited blue shift after adding organic monomers, indicating that the organic monomers may act as quenchers to extract the electrons and/or holes from the excited PbBr$_4^{2-}$ [58]. Based on our photopolymerization reaction results and mechanism studies, we proposed a mechanism of thiol-ene free-radical addition photocatalyzed by lead bromide complexes as shown in Fig. 3g. Firstly, the sulfur atoms in thiol monomers coordinate with the lead ions, while the hydrogen atoms coordinate with bromides. Upon UV light irradiation, lead bromide complexes transition from the ground state to the excited state accompanied by the generation of electrons and holes. The holes transfer to the thiol groups to generate

thiol positive radicals, which can remove the hydrogen ions immediately to generate sulfur radicals with the assistance of bromides. Finally, the lead bromide complexes that have lost holes return to the ground state under the oxidation of oxygen in the solution to complete the catalytic regeneration cycle in the aerobic atmospheric conditions. The role of oxygen as the electron scavenger was confirmed by the increasing amount of thiyl radicals in the mixture of the perovskite precursor solution and TTMP from $N_2$, to air, to $O_2$ atmosphere (Supplementary Fig. 11). In addition, the universe photocatalytic effect of lead bromide complexes was also demonstrated in other radical reactions by transforming aniline into polyaniline (Supplementary Fig. 12).

## Demonstration of PQD patterns via direct in situ photolithography

Through the direct in situ lithography method, we fabricated a series of micropatterns from mesoscale to microscale. Figure 4a shows the fluorescence microscope image of 20 μm blue stripes with high contrast between the bright blue and dark states, exhibiting clearly defined and sharp edges. The scanning electron microscope (SEM) image further demonstrates the smooth surface and high contrast between the exposed and unexposed regions in these pattern arrays (Fig. 4b). Figure 4c and Supplementary Fig. 13a show the corresponding cross-section SME image, the sharp and clean edges indicate that all the exposed line patterns were cured and the unexposed parts were removed completely. The thickness of patterns was adjustable by altering UV exposure time (Supplementary Fig. 13a), and a proportional relationship was recorded (Supplementary Fig. 13b). The inset image of the enlarged view of a single stripe (Fig. 4c) shows the thickness of patterns was more than 10 μm, which is much larger than most of the patterns within 1 μm obtained by conventional patterning processes. The absorbance spectra of the PQD-polymer films were shown in Supplementary Fig. 14, the large absorption suggests these thick films enable UV or blue light to be fully absorbed and blocked to improve energy efficiency and avoid the leakage of light[60]. We further demonstrated the luminescence uniformity of PQD patterns using a fluorescence microscope. Figure 4d depicts the emission intensity of a PQD array composed of 196 circle pixels with a diameter of 20 μm, desirable fluorescence homogeneity across all pixels can be seen. The corresponding statistical histogram of their fluorescence intensity is shown in Fig. 4e, about 70% of pixels were distributed in the range of 2000–2200, indicating the good luminescence uniformity of the PQD pixel arrays. Moreover, the radial and surface fluorescence intensity distribution of an individual 20 μm circle pixel were shown in Fig. 4f, illustrating the great fluorescence homogeneity of each pixel in PQD patterns. The 3D fluorescence image of 50 μm blue stripe patterns further confirmed the uniformity of each stripe while no residue can be seen at the unexposed regions (Supplementary Fig. 15), verifying the high efficiency and feasibility of the direct in situ photolithography.

The direct in situ photolithography can also be extended to other color PQD patterns. To make sure blue and red PPRs can be patterned, it is critical to determine the proportion of $Br^-$ substituted by $Cl^-$ or $I^-$ to ensure enough lead bromide complexes exist in the PPR. That is because both the amounts of $PbBr_4^{2-}$ and the curing rates of corresponding PPRs decreased with the increasing ratio of $[Cl^-]$ and $[I^-]$ (Supplementary Fig. 16a), and a proportional linear relationship was recorded between the curing rate and the content of $PbBr_4^{2-}$ (Supplementary Fig. 16b). To balance the ratio involving $Cl^-$ and $I^-$ to satisfy the formation of blue and red PQDs and the quantity of $PbBr_4^{2-}$ required to photocatalyze the polymerization, suitable ratios were chosen for blue and red PQDs and their catalysis capacities were confirmed by the variation of PL and PLE spectra (Supplementary Fig. 17). To obtain PQDs with good performance, annealing process was optimized for different PQDs (Supplementary Fig. 18), the climbing annealing temperature is respectively applied from blue, green and

red perovskite due to the ascending sequence lying in the solubility of precursors[61] and formation enthalpy[48]. Examples of the cartoon, letters, and the school logo composed of circle red, green, and blue pixels with a diameter and spacing of 20 μm were demonstrated in Fig. 5a–c. Colorful PQDs (Fig. 5e and Supplementary Fig. 19) were also fabricated on a substrate by using $SiO_2$ as an intermediate layer between different layers to efficiently protect the preceding layer from being destroyed by DMF and DMSO. The UV-Vis absorbance, PL emission and time-resolved PL spectra of the patterned film with stripes of 100 μm period (Fig. 5d) are shown in Supplementary Fig. 20. The peak wavelengths, PL quantum yields (QYs) and average PL lifetimes are 464 nm, 17% and 36.4 ns for blue patterns, 521 nm, 87% and 40.3 ns for green patterns, 638 nm, 58% and 61.6 ns for red patterns, respectively. A mechanism of in situ PQD fabrication can account for the good performance: a polymer matrix forms before the perovskite nucleation; the developer acts as an antisolvent to increase the supersaturation instantaneously and produce plenty of smaller nuclei[62]; further annealing of the film facilitates the growth of nuclei into larger PQDs due to the removal of solvent[63] and heat-induced diffusion of the precursor ions in the polymer matrix. The polymer matrix formed beforehand enables the controlled nucleation and growth of perovskite to facilitate a uniform PQD distribution due to the spatial confinement and the local depletion of precursors[43,64].

The higher-resolution images were created, including circle and square patterns with a size of 10 μm in Supplementary Fig. 21a and Supplementary Fig. 21b, circle patterns with a diameter of 5 μm in Fig. 5f, corresponding to the resolution of up to 2540 PPI, sufficiently enabling for the resolution of augmented reality (AR) and virtual reality (VR) display. This resolution was limited by the apparatus we used, whose minimum feature size is 5 μm, the method can be extended to fabricate higher resolution patterns by using advanced apparatus. We further evaluated the applicability of this method to various substrates which were functioned with VTMS or MPTS to enable ethenyl or thiol groups exposing, the PPR can react with these active groups to form covalent bonding under UV exposure. Clear patterns were formed not only on rigid substrates, such as glasses and wafers (Fig. 5h), but also on flexible substrates like polyethylene terephthalate (PET) (Fig. 5g, Supplementary Fig. 21c), implying broader integration of the direct in situ photolithography method. What's more, owing to no external initiators involved and effective polymer encapsulation, the fabricated PQD patterns exhibited good stability, which is highly important for further applications. The films with and without 2 w% initiators (1-hydroxycyclohexyl phenyl ketone) were evaluated with respect to UV and heat. Supplementary Fig. 22a shows the sample with initiators dropped to 7% of the original PL intensity after continuous six-day irradiation, while the one without initiators remained basically unchanged despite fluctuations. Under the heat of 60 °C, the PL intensity of the sample with initiators decreased by 82% in the first 16 h and continued dropping to 6% of the origin, while the figure for that without initiators fluctuated and remained the origin intensity until the end of test for 64 h (Supplementary Fig. 22b). In addition, at ambient temperature and normal atmosphere with an average humidity of 54%, the PL intensity of the sample without initiators can maintain 85% of the origin after 30 days (Supplementary Fig. 22c). The good stability of the PQDs might benefit from the effective encapsulation of polymer to discourage PQDs from ion migration, crystal aggregation, as well as permeation of $O_2$ and moisture[65]. By contrast, the films with external initiators may generate radicals upon irradiation, which quenched PQDs easily. Furthermore, harsher solvent tests were performed to confirm the good protection of the polymer. As can be seen in Fig. 5h and Supplementary Fig. 23a, the luminescent patterns still kept complete morphology and bright fluorescence when immersing into water and polar ethanol which can destroy PQDs easily, about 66 and 60% of the origin PL QY were retained even after 10 h (Supplementary Fig. 23b). The migration of PQDs in ethanol was determined by

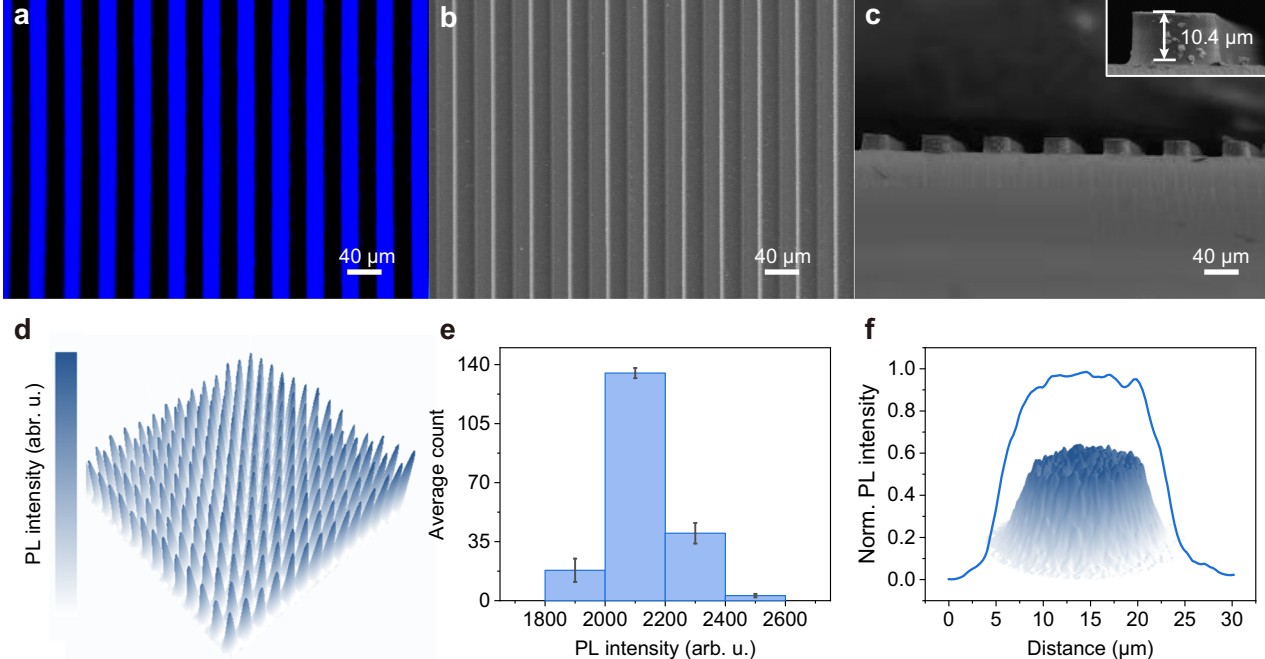

**Fig. 4 | Characterization of PQD patterns prepared via direct in situ photolithography. a** Fluorescence image, **b** SEM image, and **c** cross-section SEM image (inset image is enlarged view of a single stripe) of blue stripe patterns. **d** Emission intensity distribution image, **e** average emission intensity statistical distribution of 196 circle pixels with a diameter of 20 μm. Error bars represent standard deviation (the statistical data and errors were presented in Supplementary Table 8). **f** Radial emission intensity distribution of a single 20 μm circle pixel, inlet image is the surface-emission intensity of the pixel. Source data are provided with this paper.

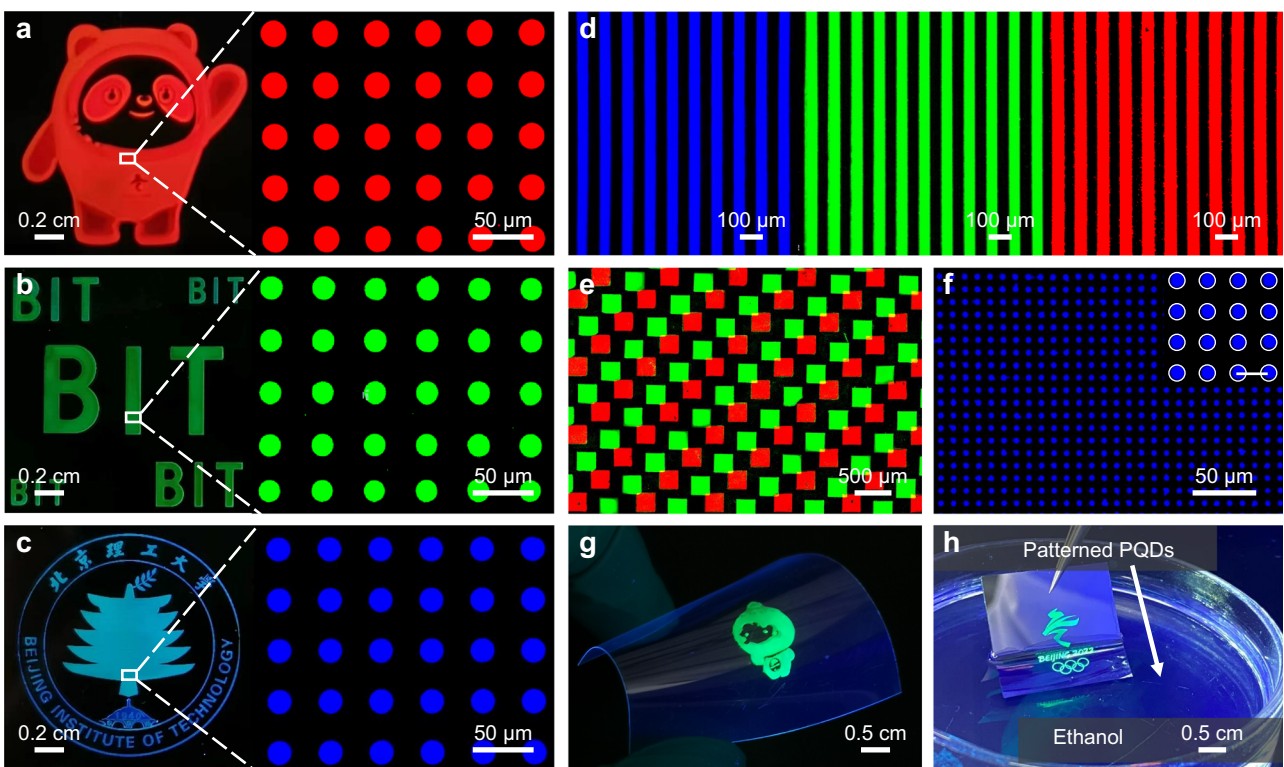

**Fig. 5 | Multicolored PQDs patterns via direct in situ photolithography.** Fluorescence images of **a** red cartoon, **b** green letters, **c** blue logo and corresponding pixels; **d** red, green and blue stripe patterns; **e** red and green double color patterned film with squares of 250 μm; **f** blue circle patterns with a diameter of 5 μm, the scale bar in the inset image is 10 μm. **g** Photograph of green cartoon on the flexible PET under UV light. **h** Photograph of green logo dipping into ethanol under UV light.

analyzing the content of Pb atoms in the film from ICP-MS and the immersed solution from ICP-OES, the proportion of 14% was assigned to the PQDs leaching from the film after 10 h immersion, illustrating the robustness of the direct in situ photolithography method. Collectively, these results demonstrate that the direct in situ photolithography technology allows for high-resolution, full-color and patterned applications.

In conclusion, by combining direct optical lithography and in situ fabrication of PQDs, we propose a non-destructive lithography method that can directly lithograph perovskite precursor solutions and then grow PQDs in situ. We demonstrate the photocatalytic role of lead bromide complexes in polymerization, thus enabling fast photolithography without the need for additional initiators or catalysts. Different from the traditional photolithography method, PQDs are generated after the lithography process, which eliminates the complex process required to prepare the PQDs in advance, at the same time effectively avoids the degradation of PQDs by solvents and high-energy UV light in the lithography process. By controlling the polymerization and in situ fabrication conditions, non-destructive efficient PQD pixel patterns can be obtained, with pixel resolution up to 2450 PPI, thicknesses up to 10 μm, and good stability to light, heat, and solvents. The direct in situ photolithography technology provides a useful idea for the preparation of non-destructive and high-efficiency light-emitting PQD pixels, and offers a simple technical route for the development of high-resolution patterned optoelectronic applications, such as Micro-LED, laser, and AR/VR devices. In the future, lead-free perovskite, including Sn-based, Bi-based, Cu-based and double perovskite, can be explored for direct in situ photolithography to avoid the use of lead. It is of great importance to develop scale-up film coating methods with high utilization of materials, such as blade, spray coating, and roll-to-roll processes.

## Methods
### Materials
All reagents were used as received without further purification: lead bromide ($PbBr_2$, 99%), lead iodide ($PbI_2$, 98%), cesium bromide (CsBr, 99.9%), cesium iodide (CsI, 99.9%), methylamine hydrobromide (MABr, 98%), methylamine hydroiodide (MAI, 98%), methylamine hydrochloride (MACl, 98%), isocyanuric acid tris(2-acryloyloxyethyl) ester (IATE, >80%) were purchased from Aladdin. 2-Phenylethanamine bromide, (PEABr, >99.5%) was purchased from Xi'an Polymer Light Technology Corp. Triallyl isocyanurate (TAIC, 98%), trimethylolpropane trimethacrylate (TMPTMA, 90%), trimethylolpropane tris(3-mercaptopropionate) (TTMP, 85%), pentaerythritol tetra(3-mercaptoproionate) (PTMP, 95%), vinyltrimethoxysilane (VTMS, 97%), 3-mercaptopropyltriethoxysilane (MPTS, 98%) were purchased from Meryer. 1,4-Butanediol bis(thioglycolate) (BBT, 98%) was purchased from Macklin. N,N-dimethylformamide (DMF, 99.5%), dimethyl sulfoxide (DMSO, ≥99.8%), trichloromethane ($CHCl_3$, ≥99%) were purchased from Beijing Tong Guang Fine Chemicals Company Co., Ltd. 5,5-Dimethyl-1pyrroline N-oxide (DMPO, 98%) was purchased from Energy-Chemical Co., Ltd.

### Preparation and characterization of PPRs
All operations were performed under atmospheric conditions. A green PPR was prepared by dissolving 0.3 mmol MABr, 0.15 mmol $PbBr_2$, 1 mmol TAIC, and 1 mmol TTMP into 0.8 mL DMF and 0.2 mL DMSO. A blue PPR was prepared by dissolving 0.4 mmol MACl, 0.2 mmol $PbBr_2$, 0.1 mmol PEABr, 1 mmol TAIC, and 1 mmol TTMP into 0.8 mL DMF and 0.2 mL DMSO. A red PPR was prepared by dissolving 0.09 mmol CsI, 0.21 mmol MAI, 0.05 mmol $PbI_2$, 0.15 mmol $PbBr_2$, 0.12 mmol PEABr 1 mmol TAIC and 1 mmol TTMP into 0.8 mL DMF and 0.2 mL DMSO. All PPRs need to be stirred for 4 h at room temperature.

FT-IR spectra were collected by Thermo Scientific™ Nicolet™ iS50 FT-IR, and the FT-IR spectrophotometer was used in conjunction with an attenuated total reflection (ATR) accessory, with zinc-selenide diamond coated plate. EPR spectra were measured by Bruker EMXplus-6/1, no chemicals were needed to detect carbon radicals, while 5,5-dimethyl-1-pyrroline-N-oxide (DMPO) was used as a radical trapping agent to detect thiyl radical. Steady-state UV–Vis absorption spectra were measured by PerkinElmer Lambda1050+, and the texting solutions were loaded in a standard quartz cuvette with an optical path of 1 mm. PLE and PL spectra were measured using an F-380 fluorescence spectrometer (Tianjin Gangdong Sci. & Tech. Development Co., Ltd.), the standard quartz cuvette with an optical path of 1 mm loaded with texting solutions was placed at a 45° angle to the light source.

### Patterning and characterization of PQDs
All operations were performed under atmospheric conditions. All substrates were cleaned in an ultrasonic bath using deionized water, ethanol, acetone, ethanol, isopropanol for 15 min each and were blown dry with a nitrogen gun. VTMS (MPTS) modifying substrates were carried out with the vapor-deposition method: a small drop of VTMS (MPTS) was dropped onto the substrates, subsequently, these substrates were heated to 80 (100) °C for 4 h. After casting 10 μL of PPR on the center of the substrate, the substrate was then bound between a black PMMA board and a patterned chrome mask. Then, a 20 W 365 nm handheld-LED was put to expose above the mask for 3–10 s for green patterns, 5–12 s for blue patterns, and 4–8 min for red patterns. After exposure, the substrate was placed on a spin coater. The uncured zone was rinsed out by spin-washing with clean chloroform, yielding micropatterns. The substrate was then annealed at 80 °C for 3 min for green patterns, room temperature for blue patterns, and 130 °C for 5 min for red patterns to remove residual solvents, and PQDs were in situ fabricated in the polymer matrix. For colorful patterns on the same substrate, an intermediate layer of $SiO_2$ was deposited between each PQD layer via vacuum magnetron sputtering.

Fluorescent images were obtained using a Nikon N-SIM A1R microscope. SEM images were taken on a ZEISS GeminiSEM. The retention and the leaching rate of the PQD-polymer films were determined by analyzing the content of Pb atom in the solvents and films based on the Agilent ICP-OES and ICP-MS 7800, and the testing samples were digested in an acidic condition and diluted by deionized water. The PL intensity of the optical images was manipulated using the camera software or ImageJ. Optical absorption spectra of patterned films were collected using a UV-6100 UV–Vis spectrophotometer (Shanghai Mapada Instruments Co., Ltd.). Photoluminescence spectra were collected with an F-380 fluorescence spectrometer (Tianjin Gangdong Sci. & Tech. Development Co., Ltd.). The ultrathin section samples were analyzed using a Tecnai G2 F30 TEM machine operating at an acceleration voltage of 300 kV. The ultrathin section samples were prepared using Leica EM UC7 ultramicrotome.

### Reporting summary
Further information on research design is available in the Nature Portfolio Reporting Summary linked to this article.

## Data availability
The data generated in this study are provided in the Source Data file. Source data are provided with this paper.

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

## Acknowledgements

The authors would like to acknowledge the Experimental Center of Advanced Materials of Beijing Institute of Technology for the support in materials synthesis and characterization. We also acknowledge Prof. He Ding, Prof. Guoguo Kang, Jianjun Li, Shangjun Cheng, Xiangang Wu, and Chenhui Wang for the useful discussion. This work was financially supported by the National Natural Science Foundation of China (62105025, G.L.Y.), Beijing Natural Science Foundation (Z210018, H.Z.Z.), National Key Research and Development Program for Young Scientists (2021YFB3601700, H.Z.Z.), National Natural Science Foundation of China (51902022, F.L.), Beijing Institute of Technology Research Fund Program for Young Scholars (3040011182113, G.L.Y.).

## Author contributions

H.Z. and G.Y. conceived and supervised the project. P.Z. fabricated the materials, conducted the method, and carried out measurements of materials. P.Z. and G.Y. analyzed data and discussed with J.S. and F.L., G.Y., and P.Z. wrote the paper, which was then discussed with H.Z.

## Competing interests

The authors declare no competing interests.
