## [Peer Review File · Nature Communications]

Direct in-situ photolithography of perovskite quantum dots via photocatalysis of lead bromide complexesREVIEWER COMMENTS

Reviewer #1 (Remarks to the Author):

The authors report the photolithography of perovskite by using perovskite precursor to photo-initiate a radical based thiol-ene polymerization reaction. The work is quite detailed, and may be considered for publication if the following comments could be addressed.

- 1) Photo-induced radical based reactions, including photo-polymerization, have been discovered a number of years back (Wong, Y.-C. et al. *Advanced Materials* 2018, 30, 1800774 and Wu, W.-B. et al. *Catalysis Science & Technology* 2018, 8, 4257). These papers include the use of perovskite to directly initiate photo-polymerization process. Due acknowledgement should be given in the background/introduction of this paper.
- 2) The authors indicated that they prepared perovskite patterns with 10um thickness for color conversion. Could the authors provide the absorbance or optical density of these films in the characterisation data? This is crucial for display applications.
- 3) Could this method be used to sequentially pattern 3 different colors of perovskite on the same substrate? Would the DMF/DMSO solvent dissolve/destroy underlying layers? Would there be halide exchange during sequential deposition of different perovskites and during developing processes? This is important for forming 3 colors on the same display substrate.
- 4) Could the authors provide the final PL quantum yield of their patterned films in the main text?
- 5) Could the authors quantify the extent of polymerization or retention of perovskites? How much monomers or perovskite is washed off during developing?
- 6) Although the authors show the Pb-S to be the active coordination site for the radical based reaction in their set of reagent, I think they should indicate that such radical reactions for perovskites do not happen exclusively for Pb-S systems. Previous works do indicate rather rich radical chemistry for perovskites, and systems without S can also perform radical based reactions.
- 7) The authors indicated the the system they used do not have initiators or catalysts that are harmful to the stability of the perovskites patterns. Do the authors have stability studies for their photo-polymerization perovskites?

Reviewer #2 (Remarks to the Author):

Patterning of PQDs is critical for the micro-optoelectronic device fabrication. In this manuscript, Zhang et al. report an in situ direct photolithography technique to pattern PQDs in the film and they reveal that the photopolymerization can be catalyzed by lead bromide complexes without addition of traditional catalyst. The mechanism has been discussed in detail. PQDs patterns are created by combining subsequent annealing, which is demonstrated to be general for patterning blue, green and red PQDs. The present technique would be compatible with the traditional optical lithography processes and enables large scale fabrication. The work is important for the patterning and applications of PQDs. I recommend it can be accepted for publication in *Nature Communications* after addressing the following concerns.

1. How about the stability of the PQDs against ethanol and UV light irradiation with the increase of the exposure time, for example over 100 hours? This information will be important for the real-world applications.
2. The PL peak of the blue PQDs is at around 464 nm. Does this originate from MAPbCl₃ PQDs? It is very interesting that no Br ions incorporate in the produced PQDs, even if the proportion of bromide is

not less than 50% in the initial precursors for triggering the polymerization. Similar unusual phenomenon happens in the patterning red PQDs. Can the authors give some comments on these phenomena? Furthermore, the Br ions or PbBr₄²⁻ complexes are always somewhere in the UV illumination domain. Do they affect the stability and emission of the PQDs after long time UV illumination that should be normal as UV light is used for the PL excitation?

3. The authors said that they fabricated colorful PQD patterns. However, the blue, green and red PQD patterns can only be generated on different substrates, not on the same one. I think that the patterns are not really "colorful". As a result, I suggest they modify the related description.

4. "3~10 s for green patterns, 5 ~ 12 s for blue patterns, and 4~8 min for red patterns" Why there is so big difference between the time for patterning blue/green PQDs and that for red PQDs?

5. Does the thickness depend on the UV light illumination time? What is the relationship?

6. Although many discussions about the photopolymerization mechanism have been given, that about the formation of PQDs are not enough. Does the PL of the PQDs depend on the annealing temperature? I suggest the authors give more information about the how they control the formation of PQDs, which is especially significant for understanding the formation of blue and red PQDs in the current work.

Reviewer #3 (Remarks to the Author):

The work reports the interesting in-situ photolithography of perovskite QD (PQD) via a photoinduced polymerization via radical-mediated thiol-ene reaction catalyzed by lead bromide complexes. The development of patterned film of PQD is intriguing and important in view of its optoelectronic application. The parameters of the PQD patterns in this work is comparable to those published recently. As highlighted in the Title of the work, the key finding in the work depends on the photocatalytic thiol-ene reaction initiated by lead bromide complexes. After a careful reading of the work, my major concern is provided below.

The precursor solution of PPR composed of all the raw materials for perovskite was first stirred for four hours at room temperature, in which the colloidal PQD should be crystallized during such a long time because of the rapid crystallization nature of MAPbBr₃. In this case, different from that discussed by the authors, the photocatalytic polymerization of thiol-ene reaction may also be initiated by the colloidal PQD. Literature survey shows a study reports recently the use of nanocrystal of metal halide perovskite as a photosensitizer to initiate thiol-ene reaction (Appl Organomet Chem. 2021; e6492.), in which the same thiol-radical nature of the reaction has been clearly illustrated. According to the reported study, it is very possible that the PQD itself serves as the photoinitiator for thiol-ene reaction in this work, but not the claimed lead bromide complexes. It is regretful that neither the PQD itself was examined as a photoinitiator, nor the abovementioned work was cited and discussed in the current study. In addition, there is also some other concern that the current study is lack of significant evidence providing a convincing identification of the lead bromide complexes, such as the anion of PbBr₄ that is proposed as the photoinitiator. Other comment, but not limited to this, includes the use of CB and VB (in figure 3 and the main text) is not reasonable as the anions of metal complexes do not have a band structure. In summary, I can not recommend the publication of the work in the pronounced journal of Nature Communications.

Reviewer #4 (Remarks to the Author):

In situ direct photolithography of perovskite quantum dots via photocatalysis of lead bromide complexes (NCOMMS-22-13723)
- Review Report -

In this manuscript the authors present a very elegant photolithographic process for patterning of PQDs in a polymer matrix, which is photocured from its thiol- and ene-monomers utilizing the PQD precursor as photoinitiator. They reveal the underlying polymerization initiation mechanism and

demonstrate impressive results achievable with their patterning method: Resolutions of 5 μm have been achieved, three colors of PQDs are accessible, the method is conducted within one step without need for purification, and resulting PQD patterns exhibit good stability due to immobilization on the surface via the polymer matrix.

PQDs are a very promising new substance class to produce color microarrays for displays due to their high efficiency, attainable colors, and low material and fabrication costs. One-step patterning processes for PQDs are indeed rare and pattern stability in general and particularly in the presence of solvents is of concern to this field. While some one-step processing techniques exist, these usually bear the disadvantage of lower resolution and lower perovskite crystal quality. Therefore, the concept of the paper is a valuable addition to ongoing research in this field.

Generally, all conclusions are supported by the presented experimental data and appropriate analysis methods have been applied. However, in some cases, additional information on presented experiments is necessary to fully prove the conclusion.

For example, the authors do not mention if oxygen is present during photocuring of the spin coated surfaces or not. Furthermore, control of layer thickness is not addressed in the manuscript. The precise parameters, under which solvent stability has been tested, are also missing in the manuscript to identify the degree of stability for PQDs. (See below for detailed questions regarding these points.)

Overall, I believe that the concept presented in this manuscript is of interest to the Nature Communications community, bridging the areas of photopolymerization and PQDs. I suggest minor revisions due to the mentioned reasons. Detailed questions regarding the scientific content are summarized below to give the authors a better understanding of the missing links for a reader of their manuscript.

Major scientific/methodic comments

1. The authors argue that the lead bromide complex regenerates via oxygen oxidation. This has not been proven by the experiments presented and furthermore, the authors do not comment whether the experiments have been performed under aerob or anearob conditions. Experiments in the presence of oxygen, in the presence of other oxidants but oxygen, and in the absence of any oxidant could prove this statement and should therefore be conducted to corroborate the authors' statement. On page 13, the authors suggest the presence of oxygen ("Finally, the lead bromide complexes that have lost holes return to the ground state under the oxidation of oxygen in the solution to complete the catalytic regeneration cycle."). If this is the case, however, how do the authors explain radical network formation without oxygen quenching?

2. Is the thickness of the pattern influenced by the coating method or curing method etc? Can it be adapted? Was it the same for all experiments? (p. 15 only mentions this matter briefly: The inset image of the enlarged view of a single stripe shows the thickness of patterns was more than 10 μm , which is much larger than most of the patterns within 1 μm obtained by conventional patterning processes.

3. Fig 4e is supposed to show average emission intensities. However, there are no error bars evident in the diagram. Precise values of the diagram, including errors, should be included as a table in the SI.

4. P. 17: "The ideal combination of direct photolithography and in situ fabrication of PQDs avoids the destruction of PQDs by solvents and high-energy UV light as PQDs were produced after lithography exposure." - Why should such good stability be given for solvents? No reasoning is explained why the final PQDs are resistant to solvents as shown in Fig S10, where PQDs obviously already exist in the polymer matrix, which could potentially migrate out of the resist structure. The conditions for these solvent tests should be added to the SI in detail (duration, were the samples at rest or in motion, weighing of samples before/after to determine potential leaching).

5. Please provide an explanation why the acrylate and methacrylate could not be polymerized (Table S2).

6. How was conversion of the polymer matrix determined (Table S4)? Were leaching experiments conducted?

Minor scientific/methodic comments

7. p. 6, line 2-3: What is meant by “stick points” and “curved products”? Naming of the chemical/physical concepts instead of these descriptive terms would be of help to readers (e.g. “covalent bonding sites for the PPR” instead of “stick points”).
8. p. 17: “Clear patterns were formed not only on rigid substrates, such as glasses and wafers, but also on flexible substrates like polyethylene terephthalate (PET), implying broader integration of the in situ direct photolithography method.” - Was covalent bonding of the PPR to the surfaces also established for these surfaces as for wafers?
9. In the concluding remarks/outlook the authors should address the weaknesses of the current system and how they may be resolved in the future (e.g. use of lead, waste of enormous amounts of PPR during spin coating process).
10. p. 18: The authors mention that high resolution of 2450 PPI could be achieved. However, they do not mention if this is the threshold resolution. Was the threshold determined?
11. p. 19: While the methods were described in sufficient detail, sample preparation was not: Were FTIR spectra measured for solutions of KBr discs? What concentrations were used for UV-vis and fluorescence measurements?
12. p. 19: Why were different annealing conditions chosen for different patterns?
13. Table/Figure descriptions (manuscript & SI): Details regarding methodic conditions under which the results represented in the corresponding Table/Figure are missing in some cases:
e.g. Fig S6: a – what solvent was used? B - what components were used and in which ratio?
Fig S7: what PPR was used, conditions for curing/developing, imaging
Fig S8: description of a-d confusing; descriptions of e and f missing
14. The review by Zou et al on Perovskite patterning from 2020 should be included: “Recent Progress on Patterning Strategies for Perovskite Light-Emitting Diodes toward a Full-Color Display Prototype”
<https://onlinelibrary.wiley.com/doi/10.1002/smssc.202000050>

Major structural comments

15. Important chemical processes are neither introduced in the introduction nor described sufficiently in the discussion or SI. This makes it difficult to follow the storyline in the beginning. For better readability more information on the curing process and the developing process should be available to readers in the beginning of the manuscript (ie introduction and/or start of results and discussion section), when photopolymerization is mentioned first.
16. Some parts of the results and discussion section (e.g. beginning) read like they should be part of the methods section and could benefit from a more concept style of reporting instead of the currently used methodic style.
e.g. “PPR is the key to the in situ direct photolithography, which is prepared by dissolving reagent salts, multifunctional thiol and ethenyl monomers in polar aprotic solvents. Specifically, the green PPR consists of MABr (MA = methylammonium), PbBr₂, trimethylolpropane tris(3-mercaptopropionate) (TTMP), triallyl isocyanurate (TAIC), N,N-dimethylformamide (DMF) and dimethyl sulfoxide (DMSO). ”
-> Instead of simply listing the components of PPR, this section would benefit from a more concept-like reporting approach, in which the components of PPR and their role are explained, with subsequent explanation why the reported chemicals were chosen from a concept point of view.

Minor structural comments

17. The beginning of the Results and discussion section would benefit from a more general description/outline of the results of the strategy instead of a description of Figure 1.
18. Abbreviation of the solvent “GBL” has not been introduced.
19. p. 16: “The in situ direct photolithography can also be extended to other color PQD patterns. To make sure blue and red PPRs can still be patterned upon UV exposure, the proportion of bromide should be no less than 50% when PPRs with different fluorescent colors were created by varying the composition of halide, more details can be found in the methods.” – proportion of bromide to which other component? Presumably other halides? Reference value is unclear in this sentence.
20. SI p. 3, Table S2: Squareroot symbol/tick symbol is misleading -> better y/n

Response to referees

We would like to express our sincere appreciation for the reviewer's constructive comments concerning our manuscript entitled "**In situ direct photolithography of perovskite quantum dots via photocatalysis of lead bromide complexes**".

These comments are very valuable and helpful for improving our manuscript. According to the reviewers' comments, we have added the requested modifications to our manuscript and supplemented extra data to improve our manuscript. In the revised version, changes to our manuscript were all highlighted within the document by using **red-colored text**.

Major modifications at a glance:

1. We add the mechanistic descriptions of the PQD formation (**Q13**) by exploring the relationship between the formation of PQDs and the annealing temperature (**Fig. S18**), and specify no PQDs were formed no matter how long the precursor solution was stirred, excluding its potential activity as photoinitiator/photosensitizer in our cause (**Q14, Fig. S7, Fig. S8, Fig. S9**).
2. The identity of the lead bromide complexes (e.g., PbBr_4^{2-}) and their role as photoinitiator species was discussed in **Q14 and Q15**, with additional absorption and PLE/PL spectra provided (**Fig. R1**). Furthermore, we demonstrate the relationship between the quantity of PbBr_4^{2-} and the curing rate of corresponding perovskite precursor resists (PPRs) (**Q11, Fig. S16**). To demonstrate the proposed regeneration via oxygen oxidation (**Q17**), we examined the amount of thiyl radicals in the N_2 -enriched, air and O_2 -enriched atmosphere (**Fig. S11**), the results show clearly that the amount of thiyl radicals increased with the increasing percentage of oxygen from N_2 , to air, to O_2 atmosphere, which can be explained by our hypothesis that O_2 works as the electron scavenger in our system and confirm the hypothesis in turn.
3. We add additional quantitative characterization including optical properties and PLQY (**Q4**), which is 58%, 87%, 17% for red, green, and blue, respectively. The extent of polymerization/retention of PQDs (**Q5**), which was determined by FI-IR spectra (**Fig. S2**), where 83% and 84% were ascribed to the conversion rate of ethenyl and thiol group, suggesting the nearly 1:1 stoichiometry reaction. Especially, we performed UV-light/solvent/heat stability measurements after photo-polymerization (**Q7**), the good stability of the photo-polymerization perovskites (**Fig. S22, Fig. S23**) might benefit from the effective encapsulation via the polymer matrix, illustrating the robustness of the in situ direct photolithography method.
4. We add additional experimental details on layer thickness control (**Q12**), the results shown in **Fig. S13** depict a proportional relationship between the thickness and the exposure time. The feasibility of sequential patterning of different colors was also demonstrated (**Q3**), new multiple-color PQD patterns on the same substrate were fabricated assisted by an intermediate layer of SiO_2 (**Fig. 5e, Fig. S19**).

The detailed responses for every concern are described as follows.

Point-by-point responses to reviewer's comments

We have carefully considered the valuable comments and suggestions from 4 reviewers, and have addressed these questions and made revisions in the revised manuscript accordingly.

Point-by-point response to Reviewer #1's comments:

The authors report the photolithography of perovskite by using perovskite precursor to photo-initiate a radical based thiol-ene polymerization reaction. The work is quite detailed, and may be considered for publication if the following comments could be addressed.

Q1 Photo-induced radical based reactions, including photo-polymerization, have been discovered a number of years back (Wong, Y.-C. et al. *Advanced Materials* 2018, 30, 1800774 and Wu, W.-B. et al. *Catalysis Science & Technology* 2018, 8, 4257). These papers include the use of perovskite to directly initiate photo-polymerization process. Due acknowledgement should be given in the background/introduction of this paper.

Response: According to your suggestion, these references and corresponding acknowledgment were added in the *Introduction* of the revised manuscript (please refer to reference 31 and 33).

Q2 The authors indicated that they prepared perovskite patterns with 10um thickness for color conversion. Could the authors provide the absorbance or optical density of these films in the characterization data? This is crucial for display applications.

Response: We agree with the reviewer's comment. The absorbance of blue, green and red PQD-polymer fabricated via in situ direct lithography was shown in Fig. S14 and discussed in Page 17. The revision is presented below (colored in blue):

Fig. S14 Absorption spectra of the PQD-polymer film. **a** Blue PQD-polymer film. **b** Green PQD-polymer film. **c** Red PQD-polymer film. All films were prepared on the VTMS modified glasses from blue, green, red PPR described in the Methods and processed follow the procedure introduced in the Methods with a 365 nm UV LED ($\sim 180 \text{ mW/cm}^2$).

Q3A Would the DMF/DMSO solvent dissolve/destroy underlying layers?

Response: Yes. DMF and DMSO solvents did destroy underlying layers. We conducted the experiment to immerse PQD-polymer composite films fabricated by our method into DMF and DMSO solvent. The result shows that the perovskite in polymer was dissolved by DMF and DMSO and thus the fluorescence of the film was quenched.

Q3B Would there be halide exchange during sequential deposition of different perovskites and during developing processes? This is important for forming 3 colors on the same display substrate.

Response: Different perovskite can dissolve the underlying layer and trigger halide exchange during sequential deposition. As shown in Fig. R1a, the coating of the blue perovskite onto the green perovskite layer induced halide exchange, resulting in the blue shift. For developing processes, because our developer didn't contain any halide ions, halide exchange wasn't observed during the developing process (Fig. R1b).

Fig. R1 PL spectra of the green PQD-polymer film fabricated by in situ direct photolithography method. **a** PL spectra of the green PQD-polymer film before and after the blue PPR deposition on it. **b** The PL spectra of green films with and without EtOAc developing.

Q3C Could this method be used to sequentially pattern 3 different colors of perovskite on the same substrate?

Response: We fabricated colorful PQD patterns on the same substrate by using SiO₂ as an intermediate layer between different colors of perovskite. These results were added in the revised manuscript (please refer to Fig. 5e and Fig. S19). The revision is presented below (colored in blue):

(Page 19 in the main text)

“Colorful PQDs (Fig. 5e and Fig. S19) were also fabricated on a substrate by using SiO₂ as an intermediate layer between different layers to efficiently protect the preceding layer from being destroyed by DMF and DMSO.”

Fig. S19 Fluorescence images of multiple color PQD patterns. **a** Red and green double color patterned film with squares of 500 μm. **b** Red, green and blue three color patterned film with squares of 250 μm. An intermediate layer of 300 nm thickness SiO₂ was deposited before patterning another emitting layer, which did not affect the optical properties of the final patterns.

Q4 Could the authors provide the final PL quantum yield of their patterned films in the main text?

Response: The PLQY of all color patterned films has been added in **Page 19** of the revised manuscript, which is 58%, 87%, 17% for red, green, and blue, respectively.

Q5 Could the authors quantify the extent of polymerization or retention of perovskites? How much monomers or perovskite is washed off during developing?

Response: We appreciate the reviewer for this important comment. The extent of polymerization was quantified by applying FT-IR measurement (as shown in **Fig. S2**). The description was added in **Page 9** of the revised manuscript. For the reservation of perovskite after developing, we performed ICP-OES and ICP-MS measurements to determine the perovskite washed off during developing by analyzing Pb atom and the results were discussed in the revised main text in **Page. 8**. The revision is presented below (colored in blue):

(Page 9 in the main text)

“The extent of polymerization was also determined by FI-IR spectra, where 83% and 84% were ascribed to the conversion rate of the ethenyl and thiol group, suggesting the nearly 1:1 stoichiometry reaction (Fig. S2).”

Fig. S2 FI-IR spectra of the PPR and cured PPR for thiol-ene polymerization analysis. a FI-IR spectra between 2670 and 2380 cm⁻¹, where the peak at 2527 cm⁻¹ represents the vibration of the thiol group. **b** FI-IR spectra between 3160 and 3020 cm⁻¹ where the peak at 3080 cm⁻¹ represents the vibration of ethenyl group. These measurements were performed by placing 50 μ L of sample on the ATR crystal plate. The UV LED (365 nm, 5 W) radiated onto the sample from a distance of 1 cm centered at the ATR crystal plate. Both peaks decayed after exposing PPR for 100 s.

(Page 8 in the main text)

“The bright and uniform PQD-polymer films can be ascribed to the good protection of polymer matrix on the exposed region, which was proved by the calculation of the retention of PQDs defined as the percentage of PQDs retained in the exposed films after developing.²⁴ Inductively coupled plasma-optical emission spectroscopy (ICP-OES) and inductively coupled plasma-mass spectroscopy (ICP-MS) were used for the analysis of Pb atom in the developer solvent and polymer film to determine the retention of PQDs. And the figure was recorded as up to 85%, which accounts for the good protection effect of polymer on perovskite thus obtaining high luminescent patterns.”

Q6 Although the authors show the Pb-S to be the active coordination site for the radical based

reaction in their set of reagent, I think they should indicate that such radical reactions for perovskites do not happen exclusively for Pb-S systems. Previous works do indicate rather rich radical chemistry for perovskites, and systems without S can also perform radical based reactions.

Response: We appreciate your constructive comments. We conducted the photopolymerization reaction by using aniline in the revised manuscript (Page 15) to explore the versatility of lead bromide complexes for various radical chemistry. As shown in Fig. S12, lead bromide complexes can also photocatalyze aniline to polymerize into polyaniline (PANI). These results suggest that lead bromide complexes are versatile to initiate radical polymerization. The revision is presented below (colored in blue):

(Page 15 in the main text)

“In addition, the universe photocatalytic effect of lead bromide complexes was also demonstrated in other radical reactions by transforming aniline into polyaniline (Fig. S12).”

Fig. S12 Exploration of the photocatalytic effect of lead bromide complexes on aniline. **a** Photograph of (i) perovskite precursor solution (PPS) under UV light illumination for 60 min; (ii) solution of aniline under UV light illumination for 60 min; (iii) PPS and aniline under sunlight for 60 min; (iv) PPS and aniline under UV light illumination for 60 min. **b** UV-vis absorption spectra of aniline solution (dot-dash line) and the mixture of PPS and aniline (solid line) in (a) under UV light for different reaction time. **c** Time-dependent absorbance profile at 435 nm extracted from (b) for tracking polyaniline formation in the presence and absence of PPS. The PPS was prepared by dissolving iso-stoichiometric CsBr and PbBr₂ with 0.05 M in DMSO; the aniline solution was prepared by blending aniline with DMSO at a volume ratio of 3:7; solutions in ii and iv were prepared by blending aniline and PPS at a volume ratio of 3:7. The 365 nm UV LED (~150 mW/cm²) was adopted to illuminate samples at the same time.

(Page 14 in the Supporting Information)

“Fig. S12a compares the samples with perovskite precursor solution (PPS) and aniline irradiated by UV light for 60 min (iv), and those without aniline (i), PPS (ii), or light (iii), separately. It is clear that the largest change of color, from light yellow to dark brown, was illustrated in the one with both perovskite precursors and aniline under UV irradiation, implying the acceleration of the reaction by applying lead bromide complexes and UV light. Furthermore, the UV spectra and the line graph (Fig. S12b, c) also confirmed the acceleration of the photocatalytic effect by lead bromide complexes. This phenomenon can be explained that the lead bromide complexes can deliver holes to the amido of aniline once exposed to UV light. The photocatalytic capacity of lead bromide complexes on aniline enables us to believe that the lead bromide complexes can work on other rich radical reactions.”

Q7 The authors indicated the system they used do not have initiators or catalysts that are harmful to the stability of the perovskites patterns. Do the authors have stability studies for their photopolymerization perovskites?

Response: As the reviewer suggested, we performed a series of stability tests. Firstly, the films with and without 2 w% initiators (1-Hydroxycyclohexyl phenyl ketone) were fabricated to test their stability with respect to UV stability and heat as shown in Fig. S22a, b. Secondly, the stability of PQD-polymer film without initiators at ambient temperature and normal atmosphere with average humidity of 54% was also recorded in Fig. S22c. Finally, the stability tests in different solvents were conducted as shown in Fig. S23. For the potential migration during the stability test in solvents, since the weight change can hardly be captured by normal scales in our laboratory, we exemplified the sample after immersing in ethanol for 10 hours and measured the migration of PQDs by analyzing the content of Pb atoms in solvents and films based on ICP-OES. All stability testing conditions, results and corresponding discussion were presented in Page 20~21 of the revised manuscript. The revision is presented below (colored in blue):

“What’s more, owing to no external initiators involved and effective polymer encapsulation, the fabricated PQD patterns exhibited outstanding stability, which is highly important for further applications. The films with and without 2 w% initiators (1-Hydroxycyclohexyl phenyl ketone) were evaluated with respect to UV and heat. Fig. S22a shows the sample with initiators dropped to 7% of the original PL intensity after continuous six-day irradiation, while the one without initiators remained basically unchanged despite fluctuations. Under the heat of 60°C, the PL intensity of the sample with initiators decreased by 82% in the first 16 h and continued dropping to 6% of the origin, while the figure for that without initiators fluctuated and remained the origin intensity until the end of test for 64 h (Fig. S22b). In addition, at ambient temperature and normal atmosphere with an average humidity of 54%, the PL intensity of the sample without initiators can maintain 85% of the origin after 30 days (Fig. S22c). The good stability of the PQDs might benefit from the effective encapsulation of polymer to discourage PQDs from ion migration, crystal aggregation, as well as permeation of O₂ and moisture.⁶⁵ By contrast, the films with external initiators may generate radicals upon irradiation, which quenched PQDs easily. Furthermore, harsher solvent tests were performed to confirm the good protection of the polymer. As can be seen in Fig. 5h and S23a, the luminescent patterns still kept complete morphology and bright fluorescence when immersing into water and polar ethanol which can destroy PQDs easily, about 66% and 60% of the origin PL QY were retained even after 10 h (Fig. S23b). The migration of PQDs in ethanol was determined by analyzing the content of Pb atoms in the film and the immersed solution from ICP-OES, the proportion of 14% was assigned to the PQDs leaching from the film after 10 h-immersion, illustrating the robustness of the in situ direct photolithography method.”

Fig. S22 Stability test of MAPbBr₃ PQD-polymer films to UV light exposure, heat and

atmosphere. **a** Remnant PL intensity for films with and without initiators versus time under 365 nm UV light irradiation ($\sim 150 \text{ mW/cm}^2$). **b** Remnant PL intensity versus heating time at 60°C . **c** Remnant PL intensity for the film without initiators versus storing time in atmosphere with average humidity of 54%. All films were prepared based on the in situ direct photolithography method with the green PPRs described in the Methods, while the films labeled “with initiators” in the graphs were deliberately introduced 2 w% initiators.

Fig. S23 Stability test for MAPbBr₃-polymer films against deterioration of solvents. **a** Fully exposed films fabricated via the in situ direct photolithography method were completely soaked at rest in the water and ethanol (EtOH) at room temperature. **b** Remnant PL QY recorded from (a).

Point-by-point response to Reviewer #2’s comments:

Patterning of PQDs is critical for the micro-optoelectronic device fabrication. In this manuscript, Zhang et al. report an in situ direct photolithography technique to pattern PQDs in the film and they reveal that the photopolymerization can be catalyzed by lead bromide complexes without addition of traditional catalyst. The mechanism has been discussed in detail. PQDs patterns are created by combining subsequent annealing, which is demonstrated to be general for patterning blue, green and red PQDs. The present technique would be compatible with the traditional optical lithography processes and enables large scale fabrication. The work is important for the patterning and applications of PQDs. I recommend it can be accepted for publication in Nature Communications after addressing the following concerns.

Q8 How about the stability of the PQDs against ethanol and UV light irradiation with the increase of the exposure time, for example over 100 hours? This information will be important for the real-world applications.

Response: We appreciate the reviewer’s valuable comment on the stability of the perovskite patterns. As this question is almost the same as Q7 from reviewer 1, please refer to the response of Q7.

Q9A The PL peak of the blue PQDs is at around 464 nm. Does this originate from MAPbCl₃ PQDs? It is very interesting that no Br ions incorporate in the produced PQDs, even if the proportion of bromide is not less than 50% in the initial precursors for triggering the polymerization. Similar unusual phenomenon happens in the patterning red PQDs. Can the authors give some comments on

these phenomena?

Response: The PL peak of the blue PQDs in our manuscript are not related to MAPbCl₃ PQDs. According to our previous work, the PL peak of MAPbCl₃ PQDs locates at 407 nm, while the MAPbCl_{0.6}Br_{2.4} presents the PL peak at 467 nm.[R1] For in situ fabricated MAPbX₃ PQDs embedded in polymer, the PL peak at 458 nm corresponds to MAPbCl_{0.5}Br_{2.5}. [R2] Based on the above data, the produced blue PQDs with a peak at 464 nm are composited of mixed Br and Cl, and Br ion has a dominant proportion, over 50%, which is similar to the ratio of Br to Cl in the used precursor solution presented in the *Methods* part. The peaks of MAPbI₃, MAPbBr_{1.5}I_{1.5} and MAPbBr_{0.9}I_{2.1} PQDs locate at 734 nm, 649 nm and 610 nm, respectively.[R1] For all-inorganic CsPbI₃ PQDs and CsPb(Br/I)₃, the typical PL peaks are at around 690 nm and 610 nm.[R3] Based on the above date, for our produced red PQDs with PL peak at 638 nm, mixed Br and I halide are also necessary.

Q9B Furthermore, the Br ions or PbBr₄²⁻ complexes are always somewhere in the UV illumination domain. Do they affect the stability and emission of the PQDs after long time UV illumination that should be normal as UV light is used for the PL excitation?

Response: According to previous reports,[R4, R5] the Br ions and PbBr₄²⁻ complexes exist in the perovskite precursor solution due to the competitive coordination between halides and solvent molecules with lead ions in the solution. For in situ fabrication of PQDs, both Br ions and PbBr₄²⁻ complexes were almost consumed during the formation of the PbBr₆⁴⁻ octahedra of perovskite.[R6] The resultant PQDs show excellent photostability against UV illumination (for stability measurement, please refer to the response of Q7).

References cited in this response:

- R1 Zhang, F. et al. Brightly Luminescent and Color Tunable Colloidal CH₃NH₃PbX₃ (X = Br, I, Cl) Quantum Dots: Potential Alternatives for Display Technology. *ACS Nano* **9**, 4533–4542, doi: 10.1021/acs.nano.5b01154 (2015).
- R2 Zhou, Q. et al. In Situ Fabrication of Halide Perovskite Nanocrystal-Embedded Polymer Composite Films with Enhanced Photoluminescence for Display Backlights. *Adv. Mater.* **28**, 9163-9168, doi:10.1002/adma.201602651 (2016).
- R3 Protesescu, L. et al. Nanocrystals of Cesium Lead Halide Perovskites (CsPbX₃, X = Cl, Br, and I): Novel Optoelectronic Materials Showing Bright Emission with Wide Color Gamut. *Nano Lett.* **15**, 3692-3696, doi:10.1021/nl5048779 (2015).
- R4 Yan, K. et al. Hybrid halide perovskite solar cell precursors: colloidal chemistry and coordination engineering behind device processing for high efficiency. *J. Am. Chem. Soc.* **137**, 4460-4468, doi:10.1021/jacs.5b00321 (2015).
- R5 Rahimnejad, S. et al. Coordination Chemistry Dictates the Structural Defects in Lead Halide Perovskites. *Chemphyschem* **17**, 2795-2798, doi:10.1002/cphc.201600575 (2016).
- R6 Shamsi, J. et al. Metal Halide Perovskite Nanocrystals: Synthesis, Post-Synthesis Modifications, and Their Optical Properties. *Chem. Rev.* **119**, 3296-3348, doi:10.1021/acs.chemrev.8b00644 (2019).

Q10 The authors said that they fabricated colorful PQD patterns. However, the blue, green and red

PQD patterns can only be generated on different substrates, not on the same one. I think that the patterns are not really “colorful”. As a result, I suggest they modify the related description.

Response: Thank you for pointing it out, in the revised manuscript, we patterned multiple color patterns on the same substrate, please refer to the response of Q3C to reviewer 1.

Q11 “3~10 s for green patterns, 5 ~ 12 s for blue patterns, and 4~8 min for red patterns” Why there is so big difference between the time for patterning blue/green PQDs and that for red PQDs?

Response: We appreciate the reviewer’s valuable comment. As we discuss in the manuscript, PbBr_4^{2-} is essential to trigger fast polymerization, so the catalytic capacity of perovskite precursor has a strong relationship with the percentage of PbBr_4^{2-} . Since the amount of PbBr_4^{2-} is positively associated with its PL intensity,[R7, R8] the PL intensity can be used to determine the amounts of PbBr_4^{2-} . The ability to form PbBr_4^{2-} varies from different solutions when different halides were used to pattern green, blue and red colors. As shown in Fig. 3f and Fig. S17, the green PPR has the maximum PL/PLE intensity, with blue PPR coming in a close second, finally is the red one with the intensity far smaller than the other two. The PbBr_4^{2-} contents reflected from PL/PLE intensity can account for the differences in exposure time, more PbBr_4^{2-} in green and blue PPR means fast patterns, while less PbBr_4^{2-} in red PPR accounts for the slow patterns.

Furthermore, to gain deep insight into the formation of PbBr_4^{2-} in halide environments and corresponding photocatalysis capacity, we conducted experiments with different ratios of $[\text{Cl}^-]$ to $[\text{Br}^-]$ to $[\text{I}^-]$ in the precursor solutions. As shown in Fig. S16a, both the amounts of PbBr_4^{2-} and the curing rates of corresponding PPRs decreased with the ratio of $[\text{Cl}^-]$ and $[\text{I}^-]$ increasing. A proportional linear relationship (Fig. S16b) was recorded between the curing rate and the content of PbBr_4^{2-} . The corresponding discussion was also added on Page18~19. The revision is presented below (colored in blue):

“To make sure blue and red PPRs can be patterned, it is critical to determine the proportion of Br substituted by Cl or I to ensure enough lead bromide complexes exist in the PPR. That is because both the amounts of PbBr_4^{2-} and the curing rates of corresponding PPRs decreased with the increasing ratio of $[\text{Cl}^-]$ and $[\text{I}^-]$ (Fig. S16a), and a proportional linear relationship was recorded between the curing rate and the content of PbBr_4^{2-} (Fig. S16b). To balance the ratio involving Cl and I to satisfy the formation of blue and red PQDs and the quantity of PbBr_4^{2-} required to photocatalyze the polymerization, suitable ratios were chosen for blue and red PQDs and their catalysis capacities were confirmed by the variation of PLE and PL spectra (Fig. S17).”

Fig. S16 The effect of replacing Br⁻ with Cl⁻ or I⁻ on the content of PbBr₄²⁻ in solutions and the curing rate of related PPRs. a The content of PbBr₄²⁻ in precursor solutions (column chart) and curing rate of corresponding PPRs (line chart) versus different ratios of [Cl⁻]:[Br⁻]:[I⁻]. **b** The curing rate versus the content of PbBr₄²⁻ extracted from (a), fitting with a straight line. The content of PbBr₄²⁻ in precursor solutions was determined by the PL intensity, and the precursor solutions were prepared by dissolving MAX and PbX₂ (n/n = 2:1) in DMSO and DMF (v/v =1:1) at the fixed lead concentration of 0.1 M. The PPRs for the curing experiment were prepared by blending equal volume of precursor solutions and the monomer solution with TTMP and TAIC (n/n = 1:1), then irradiated under 365 nm UV light to obtain the curing time, the curing rate was calculated by inverting the curing time.

References cited in this response:

- R7 Chin, S. H. et al. Tunable luminescent lead bromide complexes. *J. Mater. Chem. C* **8**, 15996-16000, doi:10.1039/d0tc04057f (2020).
- R8 Ray, A. et al. Green-emitting powders of zero-dimensional Cs₄PbBr₆: delineating the intricacies of the synthesis and the origin of photoluminescence. *Chem. Mater.* **31**, 7761-7769, doi:10.1021/acs.chemmater.9b02944 (2019).

Q12 Does the thickness depend on the UV light illumination time? What is the relationship?

Response: We appreciate the reviewer’s valuable comment. To explore the relationship between the thickness and the UV light illumination time, we fabricated the blue patterned films with different illumination time, and performed cross-section SEM measurements. The results shown in Fig. S13 depict a proportional relationship between the thickness and the exposure time. The revision is presented below (colored in blue):

(Page 16~17 in the main text)

“The thickness of patterns was adjustable by altering UV exposure time (Fig. S13a), and a proportional relationship was recorded (Fig. S13b).”

Fig. S13 Exploration of the thickness of patterns with respect to UV exposure time. a The cross-section SEM image of stripe patterns under different exposure time (2 s, 15 s, 30 s, 45 s, and 90 s). **b** The thickness of stripes versus corresponding exposure time. All patterned films with a 40 μm period were prepared on the VTMS modified glasses from a blue PPR described in the Methods and

processed following the procedure introduced in the Methods with a 365 nm UV LED (~180 mW/cm²).

Q13 Although many discussions about the photopolymerization mechanism have been given, that about the formation of PQDs are not enough. Does the PL of the PQDs depend on the annealing temperature? I suggest the authors give more information about the how they control the formation of PQDs, which is especially significant for understanding the formation of blue and red PQDs in the current work.

Response: We thank the reviewer for the important comment on the growth of PQDs. As the reviewer commented, we performed experiments to fabricate different PQD films at different annealing temperatures to explore the relationship between the formation of PQDs and the annealing temperature. The results and corresponding discussion were presented in Fig. S18 in *Supporting Information* (Page 21). In addition, we illustrated the detailed process to fabricate the PQD-polymer films and proposed the mechanism based on the exploration of the PL change versus annealing temperature in the revised manuscript (Page 19). The revision is presented below (colored in blue):

(Page 19 in the main text)

“To obtain PQDs with good performance, annealing process was optimized for different PQDs (Fig. S18), the climbing annealing temperature is respectively applied from blue, green and red perovskite due to the ascending sequence lying in the solubility of precursors⁶¹ and formation enthalpy.⁴⁸”

“A mechanism of in situ PQD fabrication can account for the good performance: a polymer matrix forms before the perovskite nucleation; the developer acts as an antisolvent to increase the supersaturation instantaneously and produce plenty of smaller nuclei;⁶² further annealing of the film facilitates the growth of nuclei into larger PQDs due to the removal of solvent⁶³ and heat-induced diffusion of the precursor ions in the polymer matrix. The polymer matrix formed beforehand enables the controlled nucleation and growth of perovskite to facilitate a uniform PQD distribution due to the spatial confinement and the local depletion of precursors.^{43, 64}”

Fig. S18 Influence of annealing temperature on PL. PL intensity of a MA_xCS_{1-x}PbI_yBr_{3-y}

polymer films, **b** MAPbBr₃-polymer films and **c** MAPbBr_xCl_{3-x}-polymer films. PL spectra of **d** MA_xCS_{1-x}PbI_yBr_{3-y}-polymer films, **e** MAPbBr₃-polymer films and **f** MAPbBr_xCl_{3-x}-polymer films. All films were fabricated via the in situ direct photolithography method with the red, green, and blue PPRs depicted in the Methods with the exception of altering annealing temperatures.

(Page 21 in the Supporting Information)

“To explore the relationship between the formation of PQDs and the annealing temperature, PQD films at different annealing temperatures were fabricated. For red perovskite, an intermediate phase tends to form after anti-solvent dripping due to interaction between Lewis base DMSO and/or iodide (I⁻) and Lewis acid PbI₂.^{2,3} The PL intensity increased with increasing annealing temperature from 90°C to 130°C (Fig. S18a), which is possibly related to the increasing conversion of the intermediate phase to the pure phase and the growth of nuclei into the larger nanocrystals. With further elevated temperature, the PL intensity presented a downward trend, which can be attributed to the thermal decomposition. Therefore, 130°C was chosen as the annealing temperature to fabricate red PQDs. By comparison, green and blue PQDs with lower solubility and formation enthalpy can be formed directly from the disordered solvate phase during solution processing,⁴ indicating lower temperature is needed. For green PQDs, the increasing PL intensity before 80°C can also be ascribed to the removal of remained solvents and the growth of PQDs. After that, a drop of the PL intensity was observed, possibly due to the aggregation of PQDs and the partial sublimation of MABr thus inducing the degradation of the PQDs. We fabricated the blue PQDs with mixed-halide of chloride/bromide. Since blue PQDs have the poorest solubility⁵ and negative formation enthalpy,⁶ they can crystallize facilely even at room temperature. In the mixed-halide perovskite, Cl⁻ and Br⁻ possess different migration rates due to the varied ion radius and binding energy, resulting in phase separation.⁷ Heat can accelerate the phase separation, consistent with the decline in the PL intensity with increasing temperature in blue PQDs (Fig. S18c).”

Point-by-point response to Reviewer #3's comments:

The work reports the interesting in-situ photolithography of perovskite QD (PQD) via a photoinduced polymerization via radical-mediated thiol-ene reaction catalyzed by lead bromide complexes. The development of patterned film of PQD is intriguing and important in view of its optoelectronic application. The parameters of the PQD patterns in this work is comparable to those published recently. As highlighted in the Title of the work, the key finding in the work depends on the photocatalytic thion-ene reaction initiated by lead bromide complexes. After a careful reading of the work, my major concern is provided below.

Q14 The precursor solution of PPR composed of all the raw materials for perovskite was first stirred for four hours at room temperature, in which the colloidal PQD should be crystallized during such a long time because of the rapid crystallization nature of MAPbBr₃. In this case, different from that discussed by the authors, the photocatalytic polymerization of thiol-ene reaction may also be initiated by the colloidal PQD. Literature survey shows a study reports recently the use of nanocrystal of metal halide perovskite as a photosensitizer to initiate thiol-ene reaction (Appl Organomet Chem. 2021; e6492.), in which the same thiol-radical nature of the reaction has been clearly illustrated. According to the reported study, it is very possible that the PQD itself serves as

the photoinitiator for thiol-ene reaction in this work, but not the claimed lead bromide complexes. It is regretful that neither the PQD itself was examined as a photoinitiator, nor the abovementioned work was cited and discussed in the current study.

Response: Thanks a lot for this valuable comment. We have investigated the origin of photopolymerization during in situ patterning PQDs by spectroscopic studies and control experiments. The key results are highlighted in the following.

(1) No PQDs were observed in our perovskite precursor resist (PPR). As shown in Fig. S7, the UV spectra of PPRs show absorbance onset at about 400 nm and the PL peaks locate around 554 nm under illumination at 365 nm. These observations can be assigned to the presence of the lead bromide complexes,[R1-R3] which are totally different from the spectroscopic features of PQDs (Fig. S7). The long-time stirring adopted in our system is just for the full dissolution of all raw materials. PPRs did not transform to PQDs even they were stirred for 450 min in our work (Fig. S7).

(2) The thiol-ene photopolymerization can be conducted without any A site ions in the PPR. To further clarify the photocatalysis of the lead bromide complexes, we conducted a comparative experiment using PbBr₂ as a photocatalyst. As shown in Table S4 and Fig. S5, the photopolymerization of TTMA and TAIC can be also processed with only PbBr₂ by prolonging the exposure time. In addition, we used the combination of HBr and PbBr₂ to replace that of MABr and PbBr₂ in the PPR, and a comparable photopolymerization rate was recorded in Fig. S9. Both of control experiments with no A site ions (MA, FA and Cs) to form PQDs but can be cured, indicating the catalysts are not PQDs.

(3) PQDs don't have the effective photocatalytic capacity as the PPR. Fig. S8 shows that all PPRs can be cured in 2 min, while the monomer system with PQDs synthesized by the hot injection method did not record any curing product even with much more irradiation time.

Furthermore, the discussion between the literature (*Appl. Organomet Chem.* 2021; 36: e6492) and our work shows lots of differences. (1) We use the perovskite precursor solution rather than formed perovskite nanocrystals as photocatalysts, they have different PL Peaks and absorption onsets (Fig. S7). (2) Different light sources were used between the literature (blue LED of 440 nm~485 nm) and our work (UV LED of 365 nm) due to the differences in absorption of the photocatalysts. (3) In the literature, they used blue LED with high power of 32 W, high conversion rate can be achieved after more than 150 min irradiation while low conversion rate of about 20% for 30 min irradiation. However, for PPRs, 84% conversion rate can be achieved in 100 s under 365 nm LED with the power of 5 W, so we exclude the photocatalytic ability of PQDs even they may catalyze thiol-ene reaction under long UV irradiation. Consequently, we assert that our results are completely different from the previous literature and the thiol-ene reaction does photocatalyze by lead bromide complexes in our case.

The revision is presented below (colored in blue):

(Page 11 in the main text)

“Since the PQDs have been reported to be used as photocatalyst for thiol-ene reactions under blue light irradiation,⁴⁰ it is easy to relate the photocatalytic effect to the PQDs. However, both the PL and UV-Vis spectra (Fig. S7) of the PPR exhibit distinctly different emission and large Stokes shift

compared with the typical PQDs, demonstrating that there was no PQD generated from PPR no matter how long the solution was stirred. All the PPRs with different stirring time can be cured within 2 min (the left five in Fig. S8), while no cured product was found in the ink using PQDs as photoinitiator (the rightest in Fig. S8), which at least indicates that PQDs cannot photocatalyze the polymerization as efficiently as the perovskite precursors. Furthermore, MABr was substituted by HBr to eliminate A site ion that is necessary for the fabrication of perovskite, avoiding any possibility to produce perovskite. The ink with HBr has a comparable curing rate to that with MABr (Fig. S9), along with the photocatalysis capacity of PbBr_2 demonstrated before, which absolutely exclude the photocatalysis effect of PQDs in our case.”

Fig. S7 Spectral studies for the PPR evolution with time. a PL and b UV-Vis spectra of the PPR with increased stirring time and the PQR ink. The PPR was prepared based on the green PPR in the Methods. The PQR ink was prepared by blending the same monomers as PPR and PQDs which were synthesized by the hot injection methods.¹

Fig. S8 Photographs of curing results of the PPR with varied stirring time and PQR ink under a sunlight and b UV light. 600 μL liquid samples were irradiated by a 365 nm UV LED (10 W) for 2 min (left five, PPR) and 4 min UV irradiation (the rightest one, PQR ink). The PPR was prepared based on the green PPR in the Methods. The PQR ink was prepared by blending the same monomers as PPR and PQDs which were synthesized by the hot injection methods.¹

Fig. S9 Curing results of different bromide sources in inks: sample (the left one) and sample (the right one) each prepared by adding 0.2 mmol MABr and HBr in 1 mL PbBr₂ solution of 0.15 M in DMSO and then mixing with equal amounts of monomers, 550 μ L of which were irradiated by a 365 nm UV light (20 W) from the same distance for 30 s.

References cited in this response:

- R1 Oldenburg, K. et al. Electronic spectra and photochemistry of tin(II), lead(II), antimony(III), and bismuth(III) bromide complexes in solution. *Z. Naturforsch. B* **48**, 1519-1523, doi: 10.1515/znb-1993-1109 (1993).
- R2 Yoon, S. J. et al. How lead halide complex chemistry dictates the composition of mixed halide perovskites. *J. Phys. Chem. Lett.* **7**, 1368-1373, doi:10.1021/acs.jpcclett.6b00433 (2016).
- R3 Chin, S. H. et al. Tunable luminescent lead bromide complexes. *J. Mater. Chem. C* **8**, 15996-16000, doi:10.1039/d0tc04057f (2020).

Q15 In addition, there is also some other concern that the current study is lack of significant evidence providing a convincing identification of the lead bromide complexes, such as the anion of PbBr₄ that is proposed as the photoinitiator.

Response: Pb²⁺ is one of the typical main group metal ions with s² electron configuration.[R1] In the perovskite precursor solution, Pb²⁺ tends to form a series of lead halide complexes. Generally, halide complexes are characterized by metal-centered sp and ligand-to-metal charge transfer (LMCT) transitions at higher energies, which can reflect on the absorption spectra.[R2] For lead bromide complexes, PbBr₂, PbBr₃⁻ and PbBr₄²⁻ have respectively been assigned to the absorption peak at about 285 nm,[R3, R4] 310 nm,[R2-R4] and 360 nm.[R2, R3] In addition, many s² complexes are photoemissive in solutions under ambient conditions because of the lowest-energy sp excited triplet.[R1] Both PbBr₃⁻ and PbBr₄²⁻ complexes are photoactive and can be featured from their emission spectra with peaks at around 600 nm and 560 nm, respectively.[R2-R5] Photoluminescence excitation (PLE) spectrum is also an effective method to analyze the species of lead bromide complexes. PLE peaks of PbBr₃⁻ and PbBr₄²⁻ correspond to about 310 nm and 360 nm.[R4, R5] UV-Vis spectra, PL spectra and PLE spectra can effectively characterize the lead bromide complexes, they were also collected in our work (Fig. 3f, Fig. S7, Fig. S10) and show good agreement with the previous reports. As shown in Fig. R1, the PPR has a strong absorbance at around 360 nm, a PL peak at 554 nm, and a PLE peak at 367 nm, which confirmed the existence of PbBr₄²⁻.

Fig. R1 Spectra of precursor solution with MABr and PbBr₂ (n/n =2:1). **a** UV-Vis spectra of 0.03 M (Pb²⁺) solution with DMF as solvent, **b** PLE with 550 nm emission and PL spectra excited at 370 nm of 0.075 M solution with DMF and DMSO (v/v=4:1) as solvents

References cited in this response:

- R1 Vogler, A. et al. Photochemistry of coordination compounds of the main group metals. *Coord. Chem. Rev.* **97**, 285-297, doi: 10.1016/0010-8545(90)80096-C (1990).
- R2 Oldenburg, K. et al. Electronic spectra and photochemistry of tin(II), lead(II), antimony(III), and bismuth(III) bromide complexes in solution. *Z. Naturforsch. B* **48**, 1519-1523, doi: 10.1515/znb-1993-1109 (1993).
- R3 Yoon, S. J. et al. How lead halide complex chemistry dictates the composition of mixed halide perovskites. *J. Phys. Chem. Lett.* **7**, 1368-1373, doi:10.1021/acs.jpcclett.6b00433 (2016).
- R4 Ray, A. et al. Green-emitting powders of zero-dimensional Cs₄PbBr₆: delineating the intricacies of the synthesis and the origin of photoluminescence. *Chem. Mater.* **31**, 7761-7769, doi:10.1021/acs.chemmater.9b02944 (2019).
- R5 Chin, S. H. et al. Tunable luminescent lead bromide complexes. *J. Mater. Chem. C* **8**, 15996-16000, doi:10.1039/d0tc04057f (2020).

Q16 Other comment, but not limited to this, includes the use of CB and VB (in figure 3 and the main text) is not reasonable as the anions of metal complexes do not have a band structure. In summary, I can not recommend the publication of the work in the pronounced journal of Nature Communications.

Response: We appreciate the reviewer for pointing out this misuse, as lead bromide complexes were formed by coordinating compounds of ions with an s² electron configuration Pb²⁺, so MO theory is more suitable for describing the electronic and geometrical structures of the metal complexes.[R10] We have corrected the “CB and VB” for lead bromide complexes as “HOMO and LOMO” in the revised manuscript (Fig. 3g).

References cited in this response:

- R10 Vogler, A. et al. The structures of s² metal complexes in the ground and sp excited states. *Comment. Inorg. Chem.* **14**, 245-261, doi:10.1080/02603599308048663 (1993).

Point-by-point response to Reviewer #4's comments:

In this manuscript the authors present a very elegant photolithographic process for patterning of PQDs in a polymer matrix, which is photocured from its thiol- and ene-monomers utilizing the PQD precursor as photoinitiator. They reveal the underlying polymerization initiation mechanism and demonstrate impressive results achievable with their patterning method: Resolutions of 5 μm have been achieved, three colors of PQDs are accessible, the method is conducted within one step without need for purification, and resulting PQD patterns exhibit good stability due to immobilization on the surface via the polymer matrix.

PQDs are a very promising new substance class to produce color microarrays for displays due to their high efficiency, attainable colors, and low material and fabrication costs. One-step patterning processes for PQDs are indeed rare and pattern stability in general and particularly in the presence of solvents is of concern to this field. While some one-step processing techniques exist, these usually bear the disadvantage of lower resolution and lower perovskite crystal quality. Therefore, the concept of the paper is a valuable addition to ongoing research in this field.

Generally, all conclusions are supported by the presented experimental data and appropriate analysis methods have been applied. However, in some cases, additional information on presented experiments is necessary to fully prove the conclusion.

For example, the authors do not mention if oxygen is present during photocuring of the spin coated surfaces or not. Furthermore, control of layer thickness is not addressed in the manuscript. The precise parameters, under which solvent stability has been tested, are also missing in the manuscript to identify the degree of stability for PQDs. (See below for detailed questions regarding these points.)

Overall, I believe that the concept presented in this manuscript is of interest to the Nature Communications community, bridging the areas of photopolymerization and PQDs. I suggest minor revisions due to the mentioned reasons. Detailed questions regarding the scientific content are summarized below to give the authors a better understanding of the missing links for a reader of their manuscript.

Major scientific/methodic comments

Q17A The authors argue that the lead bromide complex regenerates via oxygen oxidation. This has not been proven by the experiments presented and furthermore, the authors do not comment whether the experiments have been performed under aerob or anearob conditions.

Response: All experiments mentioned in the manuscript were conducted in atmospheric conditions. We have stated this point in *Results and discussion* section (Page 15) and *Methods* section (Page 22~23) in the revised manuscript (mark in red color).

Q17B On page 13, the authors suggest the presence of oxygen (“Finally, the lead bromide complexes that have lost holes return to the ground state under the oxidation of oxygen in the solution to complete the catalytic regeneration cycle.”). If this is the case, however, how do the authors explain radical network formation without oxygen quenching?

Response: PPR is quite sensitive to the content of oxygen and the effect of oxygen is complex. Oxygen can quench carbon radicals to form peroxy radicals resulting in chain termination in the polymerization.[R1] However, thiol-ene reaction is prominent for its insensitivity to oxygen, which can process easily in air. A typical radical thiol-ene reaction often starts from thiyl radicals, then

they attack double bonds to generate carbon radicals, which in turn react with thiol monomers to release thiyl radicals, resulting in the chain growth and transfer.[R2] The insensitivity to oxygen comes from that the ready hydrogen abstraction of peroxy radicals from a thiol to generate active thiyl radicals, which have the ability to continue the free-radical chain process.[R3] Hence, in our case, the radical network formation can be resistant to oxygen quenching. The thiol can reduce but not completely eliminate the quenching effect of oxygen since the carbon radical still can react with oxygen, and the reaction in air is still a little slower than that in N₂, let alone lower rate can exist in oxygen-enriched conditions.

Q17C Experiments in the presence of oxygen, in the presence of other oxidants but oxygen, and in the absence of any oxidant could prove this statement and should therefore be conducted to corroborate the authors' statement.

Response: Based on the response of Q17B, instead of observing the speed of thiol-ene reaction under the presence and absence of oxygen, we examined the amount of thiyl radicals in the N₂-enriched, air and oxygen-enriched atmosphere, as shown as in Fig. S11. The results show clearly that the amount of thiyl radicals increased with the increasing percentage of oxygen from N₂, to air, to O₂ atmosphere, which can be explained by our hypothesis that O₂ works as the electron scavenger in our system and confirm the hypothesis in turn. The revision is presented below (colored in blue):

(Page 15 in the main text)

“The role of oxygen as the electron scavenger was confirmed by the increasing amount of thiyl radicals in the mixture of the perovskite precursor solution and TTMP from N₂, to air, to O₂ atmosphere (Fig. S11).”

Fig. S11 EPR spectra for exploration of the effect of O₂. EPR spectra of perovskite precursor solution with TTMP were collected after **a** 0 min and **b** 6 min UV irradiation in the atmosphere of N₂, air and O₂. All solutions were prepared by blending green perovskite precursor solution with TTMP. The testing tubes were each inflated N₂, Air and O₂ for 10 min. 365 nm UV was used to in situ irradiate testing solutions.

References cited in this response:

- R1 Lee, T. Y., Guymon, C. A., Jönsson, E. S. & Hoyle, C. E. The effect of monomer structure on oxygen inhibition of (meth)acrylates photopolymerization. *Polymer* **45**, 6155-6162, doi:10.1016/j.polymer.2004.06.060 (2004).
- R2 Hoyle, C. E., Lee, T. Y. & Roper, T. Thiol-enes: Chemistry of the past with promise for the future. *J. Polym. Sci., Part A: Polym. Chem.* **42**, 5301-5338, doi:10.1002/pola.20366 (2004).

R3 Llorente, O. et al. Exploring the advantages of oxygen-tolerant thiol-ene polymerization over conventional acrylate free radical photopolymerization processes for pressure-sensitive adhesives. *Polym. J.* **53**, 1195-1204, doi.org/10.1038/s41428-021-00520-z (2021).

Q18 Is the thickness of the pattern influenced by the coating method or curing method etc? Can it be adapted? Was it the same for all experiments? (p. 15 only mentions this matter briefly: The inset image of the enlarged view of a single stripe shows the thickness of patterns was more than 10 μm , which is much larger than most of the patterns within 1 μm obtained by conventional patterning processes.

Response: Yes, the thickness can be adapted by altering curing time. Please refer to the response of Q12 to reviewer 2, where Fig. S13 depicts a proportional relationship between the thickness and the exposure time. Since the thickness is proportional to the curing time, for each experiment, we tried to make the curing time and the intensity of UV LED consistently to maintain the consistency in the thickness. For some special experiments, we presented the curing conditions in detail.

Q19 Fig 4e is supposed to show average emission intensities. However, there are no error bars evident in the diagram. Precise values of the diagram, including errors, should be included as a table in the SI.

Response: We have added error bars in Fig. 4e and put all precise values and errors in Table S8 in the revised *Supporting Information* (Page 17).

Q20 P. 17: “The ideal combination of direct photolithography and in situ fabrication of PQDs avoids the destruction of PQDs by solvents and high-energy UV light as PQDs were produced after lithography exposure.” - Why should such good stability be given for solvents? No reasoning is explained why the final PQDs are resistant to solvents as shown in Fig S10, where PQDs obviously already exist in the polymer matrix, which could potentially migrate out of the resist structure. The conditions for these solvent tests should be added to the SI in detail (duration, were the samples at rest or in motion, weighing of samples before/after to determine potential leaching).

Response: We appreciate the reviewer’s valuable comment. Traditional photolithography methods of PQDs are normally conducted by using preformed PQDs. These methodologies usually suffered from the degradation of PQDs under lot of solvents that are needed in photoresists, developers, etchants or strippers. In our case, PQDs are generated after the lithography process, which avoids these negative effects, we revised this sentence on Page 21. We have supplemented more stability tests, measured the migration of PQDs, and discussed the reason for such good stability in the revised manuscript (Page 20~21). Please refer to the response of Q7 to reviewer 1. In addition, the testing conditions were detailed in legends of tables/ figures regarding the testing results.

Q21 Please provide an explanation why the acrylate and methacrylate could not be polymerized (Table S2).

Response: We thank the reviewer for this careful comment. Actually, the “-” represents that the experiments we did not conduct rather than that the reaction cannot be polymerized in the original manuscript. To avoid such confusion, we modified the sign in Table S2 and added the details in

caption of Table S2 to further clarify the description. In the revised manuscript, we added more data on the photopolymerization using different monomers. The results show all combinations can be polymerized, including acrylate and methacrylate. Modification and supplement can be referred to Fig. S3c and Table S2 in the revised *Supporting Information*.

Q22 How was conversion of the polymer matrix determined (Table S4)? Were leaching experiments conducted?

Response: In Table S4, we determined the conversion of the polymer matrix by measuring the volumes of liquid residues and compared that with the original volume of the precursor inks. Although there are some deviations, this method can be applied to determine the conversion of the polymer matrix. In the revised *Supporting Information*, we added the method description in detail. Please refer to the revised legend of Table S4. Instead of leaching experiments, we quantified the extent of polymerization by FT-IR spectra. Please refer to the response of Q5.

Minor scientific/methodic comments

Q23 p. 6, line 2-3: What is meant by “stick points” and “curved products”? Naming of the chemical/physical concepts instead of these descriptive terms would be of help to readers (e.g. “covalent bonding sites for the PPR” instead of “stick points”).

Response: We appreciate the reviewer for pointing out this issue. We have corrected this sentence in Page 7 (previous Page 6). The revision is presented below (colored in blue):

“Such functionalized substrate creates strong covalent bonding sites for the resultant polymer film, which is critical to the successful patterning of perovskite films.”

Q24 p. 17: “Clear patterns were formed not only on rigid substrates, such as glasses and wafers, but also on flexible substrates like polyethylene terephthalate (PET), implying broader integration of the in situ direct photolithography method.” - Was covalent bonding of the PPR to the surfaces also established for these surfaces as for wafers?

Response: Yes, we have functionalized all substrates, including wafers and PET, with VTMS or MPTS to enable ethenyl or thiol groups exposing on the surfaces of substrates. The perovskite precursor resist (PPR) can react with these active groups to form covalent bonding under UV exposure.

Q25 In the concluding remarks/outlook the authors should address the weaknesses of the current system and how they may be resolved in the future (e.g. use of lead, waste of enormous amounts of PPR during spin coating process).

Response: As the reviewer suggested, we also discussed the weaknesses of the current system. Please refer to Page 21 in the main text. The revision is presented below (colored in blue):

“In the future, lead-free perovskite, including Sn-based, Bi-based, Cu-based and double perovskite, can be explored for in situ direct photolithography to avoid the use of lead. It is of great importance to develop scale-up film coating methods with high utilization of materials, such as blade, spray

coating and roll-to-roll processes.”

Q26 p. 18: The authors mention that high resolution of 2450 PPI could be achieved. However, they do not mention if this is the threshold resolution. Was the threshold determined?

Response: Although high resolution of 2450 PPI was achieved in the manuscript, which was limited by the apparatus we are using, whose minimum feature size is 5 μm , the method can be extended to fabricate higher resolution patterns by using advanced apparatus.

Q27 p. 19: While the methods were described in sufficient detail, sample preparation was not: Were FTIR spectra measured for solutions of KBr discs? What concentrations were used for UV-vis and fluorescence measurements?

Response: We thank the reviewer for these careful comments. A FTIR spectrophotometer (Thermo Scientific™ Nicolet™ iS50 FTIR) was used in conjunction with an attenuated total reflection (ATR) accessory, with zinc-selenide diamond coated plate. We have described the details in the *Methods* section (Page 22-23 in the main text) and figure legends (Fig. S2). For the UV-Vis and fluorescence measurement, the prepared texturing solutions were elaborated in figure legends (Fig. S7, Fig. S10, Fig. S12, Fig. S17).

Q28 p. 19: Why were different annealing conditions chosen for different patterns?

Response: We thank the reviewer for the important comment on the annealing temperature of PQDs. As this question is almost the same as Q13 from reviewer 2, please refer to the response of Q13.

Q29 Table/Figure descriptions (manuscript & SI): Details regarding methodic conditions under which the results represented in the corresponding Table/Figure are missing in some cases:

e.g. Fig S6: a – what solvent was used? B - what components were used and in which ratio?

Fig S7: what PPR was used, conditions for curing/developing, imaging

Fig S8: description of a-d confusing; descriptions of e and f missing

Response: We thank the reviewer for these careful checks. We elaborated the details in terms of methodic conditions of the results presented in all the Tables/Figures in the main text & *Supporting information* (mark in red color), especially for Fig. S10 (previous Fig. S6), Fig. S15 (previous Fig. S7) and Fig. S17 (previous Fig. S8).

Q30 The review by Zou et al on Perovskite patterning from 2020 should be included: “Recent Progress on Patterning Strategies for Perovskite Light-Emitting Diodes toward a Full-Color Display Prototype” <https://onlinelibrary.wiley.com/doi/10.1002/smsc.202000050>

Response: According to your suggestion, this reference has been cited as reference 12 in the *Introduction* section (Page 3 in the revised main text).

Major structural comments

Q31 Important chemical processes are neither introduced in the introduction nor described sufficiently in the discussion or SI. This makes it difficult to follow the storyline in the beginning.

For better readability more information on the curing process and the developing process should be available to readers in the beginning of the manuscript (ie introduction and/or start of results and discussion section), when photopolymerization is mentioned first.

Response: As the reviewer suggested, the important chemical process of the thiol-ene photopolymerization reaction was introduced in the *Introduction* section in the revised manuscript (Page 4). The revision is presented (colored in blue):

“While the thiol-ene reaction is insensitive to oxygen and high-efficiency, which has been applied in photolithography for microdevice fabrication.³⁷ A typical radical thiol-ene reaction often starts from thiyl radicals, then they attack double bonds to generate carbon radicals, which in turn react with thiol monomers to release thiyl radicals, resulting in the chain growth and transfer.³⁸ The insensitivity to oxygen comes from that the ready hydrogen abstraction of peroxy radicals from thiols to generate active thiyl radicals, which have the ability to continue the free-radical chain process.³⁹”

Q32 Some parts of the results and discussion section (e.g. beginning) read like they should be part of the methods section and could benefit from a more concept style of reporting instead of the currently used methodic style.

e.g. “PPR is the key to the in situ direct photolithography, which is prepared by dissolving reagent salts, multifunctional thiol and ethenyl monomers in polar aprotic solvents. Specifically, the green PPR consists of MABr (MA = methylammonium), PbBr₂, trimethylolpropane tris(3-mercaptopropionate) (TTMP), triallyl isocyanurate (TAIC), N,N-dimethylformamide (DMF) and dimethyl sulfoxide (DMSO).”

-> Instead of simply listing the components of PPR, this section would benefit from a more concept-like reporting approach, in which the components of PPR and their role are explained, with subsequent explanation why the reported chemicals were chosen from a concept point of view.

Response: We appreciate the reviewer for this experienced suggestion on the description of our *Results and discussion*. We rewrote these improper parts (Page 7 in the main text). The revision is presented below (colored in blue):

“The PPR is key to the in situ direct photolithography, which is prepared by dissolving reagent salts, multifunctional thiol and ethenyl monomers in polar aprotic solvents. Specifically, for green PPR, MABr (MA = methylammonium) and PbBr₂ are chosen as perovskite precursor salts, which have proved to be very effective reagents to generate perovskite. For monomers, trimethylolpropane tris(3-mercaptopropionate) (TTMP), a multi-thiol crosslinking reagent, is used as a thiol monomer, while triallyl isocyanurate (TAIC) provides electron-rich vinyl groups. To dissolve perovskite precursor and monomers better, polar aprotic solvents such as N,N-dimethylformamide (DMF) and dimethyl sulfoxide (DMSO) are the best choice.”

Minor structural comments

Q33 The beginning of the Results and discussion section would benefit from a more general description/outline of the results of the strategy instead of a description of Figure 1.

Response: According to your suggestion, we have added a more general description at the

beginning of *Results and discussion* section (Page 6 in the revised main text). The revision is presented below (colored in blue):

“Fine PQDs patterns were fabricated from the original perovskite precursor solution with monomers by in situ direct photolithography method.”

Q34 Abbreviation of the solvent “GBL” has not been introduced.

Response: We have introduced the full name of GBL (γ -butyrolactone) when it first appeared in the revised manuscript. Please refer to Page 13 in the main text (mark in red color).

Q35 p. 16: “The in situ direct photolithography can also be extended to other color PQD patterns. To make sure blue and red PPRs can still be patterned upon UV exposure, the proportion of bromide should be no less than 50% when PPRs with different fluorescent colors were created by varying the composition of halide, more details can be found in the methods.” – proportion of bromide to which other component? Presumably other halides? Reference value is unclear in this sentence.

Response: The proportion of bromide is bromide to all the halide component in a PPR. We have detailed corresponding expression, please refer to Page 18 in the revised main text (mark in red color). The revision is presented below (colored in blue):

“To make sure blue and red PPRs can be patterned, it is critical to determine the proportion of Br⁻ substituted by Cl⁻ or I⁻ to ensure enough lead bromide complexes exist in the PPR.”

Q36 SI p. 3, Table S2: Squareroot symbol/tick symbol is misleading -> better y/n

Response: We have corrected all the “squareroot symbol/tick symbol” as “y/n” in revised Table S2

We gratefully thank the reviewers for all the helpful comments which do significantly improve the quality of our manuscript.

REVIEWERS' COMMENTS

Reviewer #1 (Remarks to the Author):

The authors have addressed all the technical comments and I recommend the acceptance of the manuscript for publication.

Reviewer #2 (Remarks to the Author):

As the authors have addressed all my concerns, I recommend it can be accepted for publication by Nature Communications.

Reviewer #4 (Remarks to the Author):

Thank you very much for this thoroughly conducted revision!
I would add the response to Q24 in the manuscript and not only in the rebuttal.
I recommend the current version of the manuscript for publication in Nat Comm.
Katharina Ehrmann

Response to referees

We would like to express our sincere appreciation for the reviewer's comments concerning our manuscript entitled "**Direct in-situ photolithography of perovskite quantum dots via photocatalysis of lead bromide complexes**".

The detailed responses for every concern are described as follows.

Point-by-point responses to reviewer's comments

Point-by-point response to Reviewer #1's comments:

The authors have addressed all the technical comments and I recommend the acceptance of the manuscript for publication.

Response: We thank the reviewer for recommending our manuscript to be accepted for publication by *Nature Communications*.

Point-by-point response to Reviewer #2's comments:

As the authors have addressed all my concerns, I recommend it can be accepted for publication by Nature Communications.

Response: We thank the reviewer for recommending our manuscript to be accepted for publication by *Nature Communications*.

Point-by-point response to Reviewer #4's comments:

Thank you very much for this thoroughly conducted revision!

I would add the response to Q24 in the manuscript and not only in the rebuttal.

I recommend the current version of the manuscript for publication in Nat Comm.

Response: Thanks for the valuable comments, we have added Q24 in the revised manuscript, the revised parts were highlighted in yellow on Page 19. We appreciate your recommendation for our research to be published in *Nature Communications*.